# Aligning Protein Conformation Ensemble Generation with Physical Feedback

**Jiarui Lu** [* 1 2]  **Xiaoyin Chen** [* 1 2]  **Stephen Z. Lu** [1 3]
**Aurélie Lozano** [4]  **Vijil Chenthamarakshan** [4]  **Payel Das** [4]  **Jian Tang** [1 5 6]

## Abstract

Protein dynamics play a crucial role in protein biological functions and properties, and their traditional study typically relies on time-consuming molecular dynamics (MD) simulations conducted in silico. Recent advances in generative modeling, particularly denoising diffusion models, have enabled efficient accurate protein structure prediction and conformation sampling by learning distributions over crystallographic structures. However, effectively integrating physical supervision into these data-driven approaches remains challenging, as standard energy-based objectives often lead to intractable optimization. In this paper, we introduce Energy-based Alignment (EBA), a method that aligns generative models with feedback from physical models, efficiently calibrating them to appropriately balance conformational states based on their energy differences. Experimental results on the MD ensemble benchmark demonstrate that EBA achieves state-of-the-art performance in generating high-quality protein ensembles. By improving the physical plausibility of generated structures, our approach enhances model predictions and holds promise for applications in structural biology and drug discovery.

## 1. Introduction

Understanding protein dynamics is a critical yet complex challenge in the study of protein functionality and regulation. Protein structures transition between multiple conformational states across varying spatial and temporal scales, influencing their biological roles. Traditionally, molecular dynamics (MD) simulations have been the predominant computational tool for investigating the dynamic behavior of biological molecules. These simulations evolve Newtonian equations of motion for an entire system of particles, with accelerations determined by pre-defined force fields (physical energy functions). However, capturing biologically relevant transitions, such as folding and unfolding, often requires simulations spanning micro- to millisecond timescales (Lindorff-Larsen et al., 2011), which is computationally prohibitive, often requiring hundreds to thousands of GPU days depending on system size.

Recently, researchers have turned to deep generative models, particularly denoising diffusion models, to reframe this problem as a conditional generation task (Arts et al., 2023; Lu et al., 2024b; Jing et al., 2024a; Zheng et al., 2024). These models learn from structure databases such as the Protein Data Bank (PDB) and predict plausible conformations conditioned on specific inputs, such as amino acid sequences. While these data-driven approaches generate structurally valid candidates, they do not explicitly model thermodynamic properties. A more proper formulation of the problem is equilibrium sampling (Noé et al., 2019), which aims to sample conformation ensembles from the *Boltzmann distribution* over states. This approach is crucial for accurately modeling protein ensembles and capturing thermodynamic stability. However, it remains highly intractable, as it requires the generative models to not only produce plausible candidates but also match the underlying energy landscape. Existing amortized sampling methods (Bengio et al., 2021; Zhang & Chen, 2021; Richter & Berner, 2023; Lahlou et al., 2023) also struggle to scale to the general protein structures that typically consist of thousands of atoms.

To address these challenges, we introduce Energy-based Alignment (EBA), a framework that integrates diffusion models with physical feedback for improving protein ensemble generation. Our approach bridges the gap between purely data-driven conditional generation and physics-based simulations by incorporating fine-grained force field feedback via a scalable learning objective. Specifically, we fine-tune a pretrained all-atom denoising diffusion model to align generated conformational states with underlying physical energy landscapes. Within this alignment framework, the model interacts with force fields as an external environment, receiving feedback to refine its generative process. Through the EBA alignment, the diffusion model

---

[*]Equal contribution  [1]Mila - Québec AI Institute  [2]Université de Montréal  [3]McGill University  [4]IBM Research  [5]HEC Montréal  [6]CIFAR AI Chair. Correspondence to: Jiarui Lu <jiarui.lu@mila.quebec>, Jian Tang <jian.tang@hec.ca>.

*Proceedings of the 42$^{nd}$ International Conference on Machine Learning*, Vancouver, Canada. PMLR 267, 2025. Copyright 2025 by the author(s).

learns to balance different conformational states, resulting in more physically consistent protein conformation ensembles. We validate our approach on the ATLAS MD ensemble dataset (Vander Meersche et al., 2024) and demonstrate that the EBA-aligned diffusion model achieves state-of-the-art performance compared to previous generative models by incorporating physical feedback. The proposed method provides a novel pathway for integrating generative modeling with traditional simulations, presenting promising opportunities for protein dynamics research and drug discovery.

## 2. Preliminaries

**Notations.** A protein consisting of $L$ residues is characterized by its amino acid sequence $\mathbf{c} = (c_1, c_2, \ldots, c_L) \in |\mathcal{V}|^L$, where $\mathcal{V}$ denotes the vocabulary of 20 standard amino acids and $L$ is the number of residues. The protein structure is described by the 3D positions of its constituent atoms, $\mathbf{x} = (x_1, x_2, \ldots, x_N) \in \mathbb{R}^{N \times 3}$, encompassing all the heavy atoms (excluding hydrogen) of backbone and side-chains.

**Molecular dynamics.** Molecular dynamics (MD) simulations operate by evolving an entire particle system over a simulation time $T > 0$, governed by the physical dynamics $d\mathbf{x}_i = \mathbf{p}_i/m_i \, d\tau$, where $\mathbf{p} \in \mathbb{R}^{N \times 3}, m_i \in \mathbb{R}$ represents the momentum and the mass for each atom in the system. Given a force field (energy function) $E : \mathbb{R}^{N \times 3} \to R$, the momenta field can be updated according to the Newtonian equation of motion: $d\mathbf{p}_i = -\nabla_{\mathbf{x}_i} E(\mathbf{x}) \, d\tau$. Using an integrator like Verlet (Verlet, 1967) or Langevin dynamics (with thermostat), the position and momenta are iteratively updated at each time step, producing a simulation trajectory that converges to the Boltzmann distribution.

**Diffusion generative models.** Diffusion generative models (Sohl-Dickstein et al., 2015; Ho et al., 2020; Song et al., 2020) aim to model data distributions using a diffusion process described by a (Gaussian) Markov chain of forward distribution: $q(\mathbf{x}_t|\mathbf{x}_0, \mathbf{c}) = \mathcal{N}(\mathbf{x}_t|\alpha_t \mathbf{x}_0, \sigma_t^2 \boldsymbol{I})$, where $t \in [0, T]$ is the time step and $\alpha_t, \sigma_t > 0$ are the noise scheduling functions. The reverse (generative) process also assumes Markov structure $p_\theta(\mathbf{x}_{0:T}) = \prod_{t=1}^{T} p_\theta(\mathbf{x}_{t-1}|\mathbf{x}_t)$ and the estimated posterior can be written as (Rombach et al., 2022), $p_\theta(\mathbf{x}_{t-1}|\mathbf{x}_t) = \mathcal{N}(\mathbf{x}_{t-1}|\mu_\theta(\mathbf{x}_t, t), \sigma_{t|t-1}^2 \frac{\sigma_{t-1}^2}{\sigma_t^2} \boldsymbol{I})$, where the parameterized mean $\mu_\theta(\mathbf{x}_t, t) = \frac{\alpha_{t|t-1}\sigma_{t-1}^2}{\sigma_t^2}\mathbf{x}_t + \frac{\alpha_{t-1}\sigma_{t|t-1}^2}{\sigma_t^2}\mathbf{x}_\theta(\mathbf{x}_t, t)$, $\sigma_{t|s}^2 = \sigma_t^2 - \alpha_{t|s}^2 \sigma_s^2$ and $\alpha_{t|s} = \alpha_t/\alpha_s$ for $s < t$. Following Ho et al. (2020), the denoising objective can be expressed as:

$$\mathcal{L} = \mathbb{E}_{\mathbf{x}_0 \sim D} \mathbb{E}_{t \sim \mathcal{U}(0,T), \mathbf{x}_t \sim q(\mathbf{x}_t|\mathbf{x}_0)} \left[ \omega(\lambda_t) \|\epsilon - \epsilon_\theta(\mathbf{x}_t, t)\|_2^2 \right], \quad (1)$$

where $\epsilon \sim \mathcal{N}(0, \boldsymbol{I})$, $\lambda_t \triangleq \alpha_t^2/\sigma_t^2$ is a signal-to-noise ratio (SNR) and $\omega > 0$ is the loss reweighting function.

**Reinforcement learning from human feedback.** Reinforcement Learning from Human Feedback (RLHF) represents a canonical example of the alignment problem. The goal of RLHF is to align a generative model $p_\theta(\mathbf{x}|\mathbf{c})$ to maximize the reward $r_\theta(\mathbf{x}, \mathbf{c})$ which is provided by a model parameterized by $\theta$, while being constrained by a reference distribution $p_{\text{ref}}(\mathbf{x}|\mathbf{c})$ ($\alpha > 0$ is the regularization factor):

$$\max_{p_\theta} \mathbb{E}_{p_\theta(\mathbf{x})} \left[ r_\theta(\mathbf{x}, \mathbf{c}) \right] - \alpha \mathbb{D}_{\text{KL}} \left( p_\theta(\mathbf{x}|\mathbf{c}) || p_{\text{ref}}(\mathbf{x}|\mathbf{c}) \right). \quad (2)$$

Prior work (Peters & Schaal, 2007; Peng et al., 2019; Rafailov et al., 2024) has shown that this objective has a closed-form solution:

$$p_\theta(\mathbf{x}|\mathbf{c}) = \frac{1}{Z} p_{\text{ref}}(\mathbf{x}|\mathbf{c}) e^{r_\theta(\mathbf{x},\mathbf{c})/\alpha}, \quad (3)$$

where $Z = \sum_{\mathbf{x}} p_{\text{ref}}(\mathbf{x}|\mathbf{c}) e^{\frac{r(\mathbf{x},\mathbf{c})}{\alpha}}$ is the partition function. Accordingly, the reward function can be written as:

$$r_\theta(\mathbf{x}, \mathbf{c}) = \alpha \log \frac{p_\theta(\mathbf{x}|\mathbf{c})}{p_{\text{ref}}(\mathbf{x}|\mathbf{c})} + \alpha \log Z. \quad (4)$$

Direct Preference Optimization (DPO) (Rafailov et al., 2024) has demonstrated that when the reward model is optimized using the Bradley-Terry (BT) model $p(\mathbf{x}^w \succ \mathbf{x}^l|\mathbf{c}) = \sigma(r_\theta(\mathbf{x}^w, \mathbf{c}) - r_\theta(\mathbf{x}^l, \mathbf{c}))$, the RL problem in Eq. (2) is simplified to supervised learning on pairwise preference data:

$$L_{\text{DPO}}(\theta) = -\mathbb{E}_{(\mathbf{c}, \mathbf{x}^w, \mathbf{x}^l) \sim D}$$
$$\log \sigma \left( \alpha \log \frac{p_\theta(\mathbf{x}^w|\mathbf{c})}{p_{\text{ref}}(\mathbf{x}^w|\mathbf{c})} - \alpha \log \frac{p_\theta(\mathbf{x}^l|\mathbf{c})}{p_{\text{ref}}(\mathbf{x}^l|\mathbf{c})} \right), \quad (5)$$

where $\sigma(\cdot)$ is the sigmoid function.

## 3. Method

In this section, we introduce Energy-based Alignment (EBA), a novel physics-informed learning objective for improving conformation ensemble generation. EBA leverages the underlying Boltzmann distribution by aligning a model's learned distribution with energy-induced probability ratios, while circumventing the need to explicitly compute the intractable partition function. Instead, it employs an energy-weighted classification-like objective over a stochastic mini-batch combination of conformational states, ensuring that physically meaningful energy terms directly guide the alignment process. We further demonstrate the connection between EBA and DPO, providing an alternative view of the alignment problem our method optimizes.

### 3.1. Energy-based Alignment

Given an amino acid sequence (or multiple sequence alignment, MSA) as condition $\mathbf{c}$, our goal is to conditionally sample the protein conformation ensemble $\{\mathbf{x}^i\}$ that belongs to $\mathbf{c}$ from the target Boltzmann distribution induced

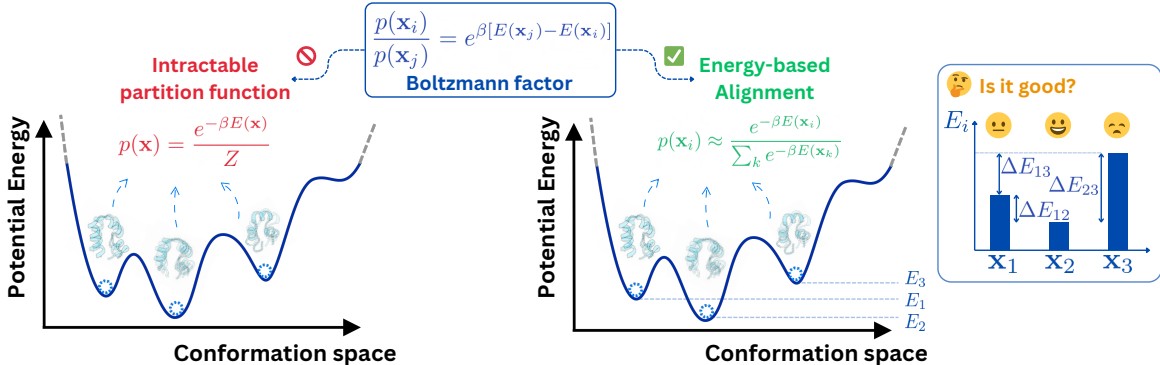

*Figure 1.* Motivation behind the proposed Energy-based Alignment (EBA). To align the generative model to adhere to the *Boltzmann factor* across different conformational states, we employ a stochastic approximation of the partition function which simplifies the intractable $Z$ over all possible states to a finite summation, enabling efficient learning from physical energy while maintaining training scalability.

by some potential energy function $E(\mathbf{x}; \mathbf{c}) \in \mathbb{R}$, that is, $p_B(\mathbf{x}|\mathbf{c}) = e^{-\beta E(\mathbf{x}; \mathbf{c})}/Z$, where $\beta \triangleq 1/k_B T$ is the temperature factor and $Z \triangleq \sum_{\mathbf{x}} e^{-\beta E(\mathbf{x}; \mathbf{c})}$. However, directly sampling from such distribution is intractable due to the non-trivial partition function (or normalizing constant) $Z \triangleq \sum_{\mathbf{x}} e^{-\beta E(\mathbf{x}; \mathbf{c})}$ which requires the sum (or integration) over all possible conformation states in the high-dimension space. An relevant concept is the *Boltzmann factor*, which describes the ratio of probabilities of two states $\mathbf{x}^i$ and $\mathbf{x}^j$ as a function of energy difference: $\frac{p_B(\mathbf{x}^i|\mathbf{c})}{p_B(\mathbf{x}^j|\mathbf{c})} = e^{-\beta \Delta E_{ij}}$ where $\Delta E_{ij} = E(\mathbf{x}^i; \mathbf{c}) - E(\mathbf{x}^j; \mathbf{c})$. This dependence on $\Delta E_{ij}$ ensures that the Boltzmann factor is invariant to shifts in absolute energy values, which is particularly important for generative model training since energy scales can vary significantly with the number of atoms in the protein. Leveraging $\Delta E_{ij}$ enables us to construct learning objectives that effectively incorporate energy feedback, capturing the relative stability of states within the conformation ensemble.

Suppose that $p_\theta(\mathbf{x}|\mathbf{c}) = e^{-\alpha E_\theta(\mathbf{x}; \mathbf{c})}/Z(\alpha > 0)$ is a learnable probabilistic model parameterized by $E_\theta$, then we can optimize the model by minimizing the Kullback-Leibler (KL) divergence against the target Boltzmann distribution via the cross-entropy:

$$\mathbb{D}_{\mathrm{KL}}(p_B(\mathbf{x}|\mathbf{c}) \| p_\theta(\mathbf{x}|\mathbf{c})) = \sum_i p_B(\mathbf{x}^i|\mathbf{c}) \log \left( \frac{p_B(\mathbf{x}^i|\mathbf{c})}{p_\theta(\mathbf{x}^i|\mathbf{c})} \right)$$

$$= \sum_i p_B(\mathbf{x}^i|\mathbf{c}) \log p_B(\mathbf{x}^i|\mathbf{c}) - \sum_i p_B(\mathbf{x}^i|\mathbf{c}) \log p_\theta(\mathbf{x}^i|\mathbf{c})$$

$$= -\sum_i p_B(\mathbf{x}^i|\mathbf{c}) \log p_\theta(\mathbf{x}^i|\mathbf{c}) - H(p_B; \mathbf{c})$$

$$= -\sum_i \frac{e^{-\beta E(\mathbf{x}^i; \mathbf{c})}}{\sum_j e^{-\beta E(\mathbf{x}^j; \mathbf{c})}} \log \left( \frac{e^{-\alpha E_\theta(\mathbf{x}^i; \mathbf{c})}}{\sum_j e^{-\alpha E_\theta(\mathbf{x}^j; \mathbf{c})}} \right) + Const, \tag{6}$$

where $H(p_B; \mathbf{c}) \equiv -\sum_i p_B(\mathbf{x}^i|\mathbf{c}) \log p_B(\mathbf{x}^i|\mathbf{c})$ is the en-

tropy of the target distribution which is constant w.r.t. $\theta$. Direct optimization of Eq. (6) is intractable due to the summation over all possible conformational states. We approximate it by considering a stochastic finite subset of $K$ representative states $\{\mathbf{x}^i\}_{i=1}^K$ from some proposal distribution $p^*$, such as reference MD simulations or generative models. Then we rewrite the first term in Eq. (6) as:

$$\mathcal{L}_{\mathrm{EBA}}(\theta) = -\mathbb{E}_{(\mathbf{c}, \{\mathbf{x}^i\}) \sim p^*}$$
$$\left[ \sum_{i=1}^K \frac{e^{-\beta E(\mathbf{x}^i; \mathbf{c})}}{\sum_{j=1}^K e^{-\beta E(\mathbf{x}^j; \mathbf{c})}} \log \frac{e^{-\alpha E_\theta(\mathbf{x}^i; \mathbf{c})}}{\sum_{j=1}^K e^{-\alpha E_\theta(\mathbf{x}^j; \mathbf{c})}} \right], \tag{7}$$

which we refer to as the Energy-based Alignment (EBA) objective. EBA enables stochastic optimization via mini-batch sampling, where the finite-state approximation in Eq. (7) preserves the *Boltzmann factor* within each mini-batch. This ensures that the probability ratio between any two states $\mathbf{x}^i$ and $\mathbf{x}^j$, remains proportional to their energy difference, i.e., $\frac{p_\theta(\mathbf{x}^i|\mathbf{c})}{p_\theta(\mathbf{x}^j|\mathbf{c})} = e^{-\alpha \Delta E_{ij}}$. Consequently, EBA approximately aligns the model's learned distribution with the target Boltzmann distribution while preserving the probability ratio.

### 3.2. EBA for Diffusion Models

Eq. (7) presents a general form where $E_\theta(\mathbf{x}; \mathbf{c})$ is not specifically defined. To adapt this objective for diffusion models, we stipulate $E_\theta(\mathbf{x}; \mathbf{c})$ in terms of the probability distribution $p_\theta(\mathbf{x}|\mathbf{c})$. Suppose that the diffusion model is specified with forward $q(\mathbf{x}_{1:T}|\mathbf{x}_0)$ and parameterized $p_\theta(\mathbf{x}_{0:T})$. We here explore an efficient form of objective for diffusion models under the EBA framework as follows. To begin with, we consider the basic form of energy function by using the negative log-likelihood: $E_\theta = -\log p_\theta(\mathbf{x}|\mathbf{c})$, where the $\log Z$ is omitted because it is constant w.r.t. $\mathbf{x}$. For diffusion models, there exist a chain of latents $\mathbf{x}_{1:T}$ and we alter the energy to be on the whole chain: $-\mathbb{E}_{p_\theta(\mathbf{x}_{1:T}|\mathbf{c})} \log p_\theta(\mathbf{x}_{0:T}|\mathbf{c})$. Then we can factorize the

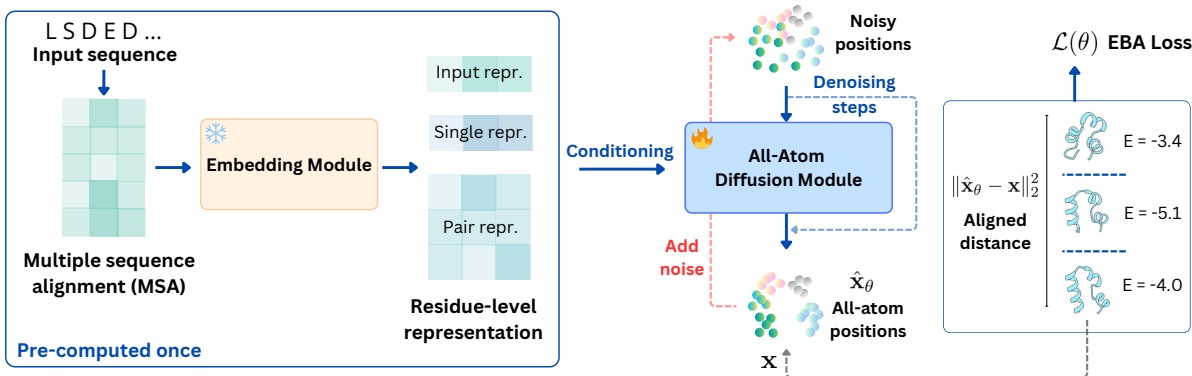

*Figure 2.* Illustration of model architecture and EBA pipeline for energy-weighted preference fine-tuning. Starting from an input sequence and its MSA, residue-level embeddings are pre-computed and conditioned to guide the all-atom diffusion. At each step, $K$ reference structures (e.g., $K = 3$) with the same sequence are sampled from the dataset, and their energies $E$ are evaluated. The denoising error is measured by aligned distance as in Eq. (14), and combined with $E$ into the EBA-diffusion loss.

joint distribution with the reverse decomposition for $p_\theta$, i.e., $-p_\theta(\mathbf{x}_T) - \sum_{t=1}^{T} \mathbb{E}_{p_\theta(\mathbf{x}_t, \mathbf{x}_{t-1}|\mathbf{x}_0, \mathbf{c})} \log p_\theta(\mathbf{x}_{t-1}|\mathbf{x}_t, \mathbf{c})$. Since the prior $p_\theta(\mathbf{x}_T)$ is typically chosen as a standard Gaussian, the first term becomes a constant. However, to optimize the EBA objective containing the above decomposition, we have to sample the latents according to $p_\theta(\mathbf{x}_t, \mathbf{x}_{t-1}|\mathbf{x}_0, \mathbf{c})$, which in practice is intractable because of the expensive simulation of the reverse process. Therefore, we substitute the reverse process of $p_\theta(\mathbf{x}_{1:T}|\mathbf{x}_0)$ with the forward kernel (Wallace et al., 2024), and the energy function now becomes $-\sum_{t=1}^{T} \mathbb{E}_{q(\mathbf{x}_t|\mathbf{x}_0, \mathbf{c})} \mathbb{E}_{q(\mathbf{x}_{t-1}|\mathbf{x}_{0,t})} \log p_\theta(\mathbf{x}_{t-1}|\mathbf{x}_t, \mathbf{c})$. Due to the introduction of forward $q(\cdot)$, we amend with an entropy term and have the final definition of energy function:

$$E_\theta(\mathbf{x}; \mathbf{c}) \triangleq$$
$$-\sum_{t=1}^{T} \mathbb{E}_{q(\mathbf{x}_t|\mathbf{x}_0)} \left( \mathbb{E}_{q(\mathbf{x}_{t-1}|\mathbf{x}_{0,t})}[\log p_\theta(\mathbf{x}_{t-1}|\mathbf{x}_t, \mathbf{c})] + H_t(q) \right)$$
$$= \sum_{t=1}^{T} \mathbb{E}_{q(\mathbf{x}_t|\mathbf{x}_0)} \left( \mathbb{D}_{\mathrm{KL}}[q(\mathbf{x}_{t-1}|\mathbf{x}_{0,t}) \| p_\theta(\mathbf{x}_{t-1}|\mathbf{x}_t, \mathbf{c})] \right) \quad (8)$$

where $H_t(q) \triangleq -\mathbb{E}_{q(\mathbf{x}_{t-1}|\mathbf{x}_{0,t})} q(\mathbf{x}_{t-1}|\mathbf{x}_{0,t})$ denotes the (self) entropy of posterior defined by forward kernel. Intuitively, the energy function is defined by the KL-divergence terms according to the decomposition of the reverse process.

By plugging Eq. (8) back into Eq. (7), we further leverage the fact that the log-sum-exp (LSE) function is convex and therefore apply Jensen's inequality to push out the expectation (summation), leading to the following upper bound after some algebra (see Appendix A.1 for detailed derivation):

$$\mathcal{L}(\theta) \leq -\mathbb{E}_{(c,\{\mathbf{x}_0^i\}_{i=1}^K)\sim D, t\sim\mathcal{U}(0,T), \mathbf{x}_t^i\sim q(\mathbf{x}_t^i|\mathbf{x}_0^i)}$$
$$\sum_{i=1}^{K} \frac{e^{-\beta E(\mathbf{x}^i;\mathbf{c})}}{\sum_{j=1}^{K} e^{-\beta E(\mathbf{x}^j;\mathbf{c})}} \log \frac{e^{-\alpha T \mathbb{D}_{\mathrm{KL}}[q(\mathbf{x}_{t-1}^i|\mathbf{x}_{0,t}^i)\|p_\theta(\mathbf{x}_{t-1}^i|\mathbf{x}_t^i,\mathbf{c})]}}{\sum_{j}^{K} e^{-\alpha T \mathbb{D}_{\mathrm{KL}}[q(\mathbf{x}_{t-1}^j|\mathbf{x}_{0,t}^j)\|p_\theta(\mathbf{x}_{t-1}^j|\mathbf{x}_t^j,\mathbf{c})]}}$$
$$(9)$$

Because both $q(\mathbf{x}_{t-1}|\mathbf{x}_{0,t})$ and $p_\theta(\mathbf{x}_{t-1}|\mathbf{x}_t, \mathbf{c})$ can be parameterized by gaussian distribution, we thus substitute the KL divergence with the denoising term, setting the weighting function $\omega(\lambda_t)$ as constant, we obtain our EBA objective for diffusion as follows:

$$\mathcal{L}_{\text{EBA-Diffusion}}(\theta) = -\mathbb{E}_{(c,\{\mathbf{x}\}_{i=1}^K)\sim D, t\sim\mathcal{U}(0,T), \mathbf{x}_t^i\sim q(\mathbf{x}_t^i|\mathbf{x}_0^i)}$$
$$\sum_{i=1}^{K} \frac{e^{-\beta E(\mathbf{x}^i;\mathbf{c})}}{\sum_{j=1}^{K} e^{-\beta E(\mathbf{x}^j;\mathbf{c})}} \log \frac{e^{-\alpha T(\|\epsilon^i - \epsilon_\theta(\mathbf{x}_t^i, t, \mathbf{c})\|_2^2)}}{\sum_{j=1}^{K} e^{-\alpha T(\|\epsilon^j - \epsilon_\theta(\mathbf{x}_t^j, t, \mathbf{c})\|_2^2)}}.$$
$$(10)$$

### 3.3. DPO as a Special Case of EBA

Note that the derivation of the EBA objective in Eq. (7) is independent of the form of energy function $E_\theta(\mathbf{x}; \mathbf{c})$. A notable alternative parametrization is the negative log-likelihood ratio between the training model $p_\theta(\mathbf{x}|\mathbf{c})$ and a reference model $p_{\text{ref}}(\mathbf{x}|\mathbf{c})$ (e.g., a frozen checkpoint of the pre-trained model): $E_\theta(\mathbf{x}; \mathbf{c}) = -\log \frac{p_\theta(\mathbf{x}|\mathbf{c})}{p_{\text{ref}}(\mathbf{x}|\mathbf{c})}$. Using this parametrization, DPO emerges as a special case of EBA when $K = 2$ and low temperature (i.e., $\beta \to \infty$). Without loss of generality, consider a pair of samples

$\mathbf{x}_1, \mathbf{x}_2 \; s.t. \; E(\mathbf{x}_1, \mathbf{c}) < E(\mathbf{x}_2, \mathbf{c})$:

$$\mathcal{L}_{\text{EBA-DPO}}(\theta) =$$

$$- \mathbb{E}_{(\mathbf{c}, \mathbf{x}^1, \mathbf{x}^2)} \sum_{i=1}^{2} \frac{e^{-\beta E(\mathbf{x}^i; \mathbf{c})}}{\sum_{j=1}^{2} e^{-\beta E(\mathbf{x}^j; \mathbf{c})}} \log \frac{e^{\alpha \log \frac{p_\theta(\mathbf{x}^i|\mathbf{c})}{p_{\text{ref}}(\mathbf{x}^i|\mathbf{c})}}}{\sum_{j=1}^{2} e^{\alpha \log \frac{p_\theta(\mathbf{x}^j|\mathbf{c})}{p_{\text{ref}}(\mathbf{x}^j|\mathbf{c})}}}$$

$$= -\mathbb{E}_{(\mathbf{c}, \mathbf{x}^1, \mathbf{x}^2)} \log \frac{e^{\alpha \log \frac{p_\theta(\mathbf{x}^1|\mathbf{c})}{p_{\text{ref}}(\mathbf{x}^1|\mathbf{c})}}}{e^{\alpha \log \frac{p_\theta(\mathbf{x}^1|\mathbf{c})}{p_{\text{ref}}(\mathbf{x}^1|\mathbf{c})}} + e^{\alpha \log \frac{p_\theta(\mathbf{x}^2|\mathbf{c})}{p_{\text{ref}}(\mathbf{x}^2|\mathbf{c})}}}$$

$$(\beta \to \infty)$$

$$= -\mathbb{E}_{(\mathbf{c}, \mathbf{x}^1, \mathbf{x}^2)} \log \sigma \left( \alpha \log \frac{p_\theta(\mathbf{x}^1|\mathbf{c})}{p_{\text{ref}}(\mathbf{x}^1|\mathbf{c})} - \alpha \log \frac{p_\theta(\mathbf{x}^2|\mathbf{c})}{p_{\text{ref}}(\mathbf{x}^2|\mathbf{c})} \right),$$

$$(11)$$

which is equivalent to Eq. (5). This derivation demonstrates that EBA reduces to a DPO objective when sampling two states and ignoring energy differences between states. Following Wallace et al. (2024), we also derive the DPO variation of EBA for diffusion models in Appendix A.2.

---

**Algorithm 1** Fine-tuning Diffusion Model with EBA

---

**Require:** Pre-trained denoising network $\mathbf{x}_\theta$, conformation dataset $D = \{(\mathbf{c}, \mathbf{x}_0)\}$ with sequence conditions $\mathbf{c}$ and corresponding structures $\mathbf{x}_0$, energy function $E(\mathbf{x}; \mathbf{c})$, inverse temperature factors $\alpha, \beta$, the number of time steps $T$, learning rate $\gamma$.

1: **while** not converged **do**
2:     Sample $(\mathbf{c}, \mathbf{x}_0)$ from $D$
3:     Let $\mathbf{x}_0^1 \leftarrow \mathbf{x}_0$
4:     Randomly retrieve $K-1$ samples $(\mathbf{c}^i, \mathbf{x}_0^i) \sim D$ s.t. $\mathbf{c}^i = \mathbf{c} \; (i = 2, \ldots, K)$
5:     Calculate energy $E(\mathbf{x}_0^i; c)$ for each $i = 1, \ldots, K$
6:     Calculate mini-batch Boltzmann weights:

$$w(\mathbf{x}_0^i) = \frac{e^{-\beta E(\mathbf{x}_0^i; c)}}{\sum_{j=1}^{K} e^{-\beta E(\mathbf{x}_0^j; c)}}$$

7:     **for** each candidate $i = 1, \ldots, K$ **do**
8:         Sample timestep $t \sim \text{Uniform}(0, T)$
9:         Add noise $\mathbf{x}_t^i \sim q(\mathbf{x}_t^i | \mathbf{x}_0^i)$
10:        Forward denoising network $\mathbf{x}_\theta(\mathbf{x}_t^i, t)$
11:        Calculate the aligned loss $\mathcal{L}_{\text{total}}^i(\theta)$ in Eq. (14)
12:    **end for**
13:    Calculate EBA loss according to Eq. (10)

$$\mathcal{L}_{\text{EBA-Diffusion}} = -\sum_{i=1}^{K} w(\mathbf{x}_0^i) \log \frac{e^{-\mathcal{L}_{\text{total}}^i}}{\sum_{j=1}^{K} e^{-\mathcal{L}_{\text{total}}^j}}$$

14:    Update parameters: $\theta \leftarrow \theta - \gamma \nabla_\theta \mathcal{L}_{\text{EBA-Diffusion}}$
15: **end while**
16: **Return** fine-tuned denosing network $\mathbf{x}_\theta$.

---

## 3.4. All-atom Diffusion with AlphaFold

We base our conditional generative model on the AlphaFold3 architecture (Abramson et al., 2024), which encodes sequence information through its MSA Module and PairFormer into conditioning embeddings, followed by an all-atom diffusion module to iteratively refine atomic coordinates from a noise distribution into clean conformations. AlphaFold3 explicitly models atom coordinates in the protein structure, rather than relying on coarse-grained rigid frames for backbone atoms and internal torsion angles for side chain atoms which can be hindered to capture subtle structural variations and alternative conformations. Adopting all-atom modeling architecture enables a more direct representation of conformational degrees of freedom and facilitates accurate integration of physical feedback.

## 3.5. Structural Alignment-based Objective

From the derivation of objective in Eq. (10), the standard mean squared error (MSE) or L2 loss $L(\theta) = \|\epsilon - \epsilon_\theta(\mathbf{x}_t^j, t)\|_2^2$ is commonly used to minimize the difference between predictions and ground truth. However, this MSE term may be suboptimal for protein conformation generation. The input condition (amino acid sequences) is SE(3)-invariant[1], meaning it does not differentiate between roto-translational variants of the target protein, which can negatively affect the training efficiency. Therefore, we consider the following SE(3)-invariant losses for EBA fine-tuning:

**Rigid aligned MSE.** To ensure that the loss remains invariant to global rotational and translational differences, we first align the predicted atomic coordinates to the ground truth structure using the Kabsch's algorithm. This alignment minimizes the root-mean-square deviation (RMSD) by applying an optimal rotation and translation. The MSE loss is then computed on the aligned coordinates of all atoms, allowing the model focuses on meaningful conformational changes rather than arbitrary positional offsets:

$$L_{\text{Aligned MSE}} = \frac{1}{N} \sum_{l=1}^{N} \|\text{Align}(\hat{\mathbf{x}}_0, \mathbf{x}_0)_l - \mathbf{x}_{0,l}\|_2^2, \quad (12)$$

where $\text{Align}(\hat{\mathbf{x}}_0, \mathbf{x}_0)$, aligns the predicted coordinates $\hat{\mathbf{x}}_0$ to the ground truth coordinates $\mathbf{x}_0 \in \mathbb{R}^{N \times 3}$, and $N$ is the number of atoms in the structure ($l = 1, \ldots, N$).

**Smooth LDDT.** While alignment-based MSE ensures structural consistency in Cartesian space, it does not explicitly capture inter-atomic relationships that are important for protein geometry. To address this, we additional introduce an auxiliary loss term based on the pairwise distances.

---

[1]A function $f$ is said to be SE(3)-invariant if its output remains unchanged under arbitrary rigid transformations (i.e., rotations and translations): $f(R\mathbf{x} + t) = f(\mathbf{x}), \forall R \in \text{SO}(3), \; t \in \mathbb{R}^3$.

*Table 1.* Statistical metrics on MD ensemble benchmark of ATLAS test set ($N = 250$ following Jing et al. (2024a)) where the median across all test targets is reported. The runtime is reported as **GPU second** required per sample averaged on all test targets. The best result is highlighted in **bold**, while the second-best result is underlined.

| Metrics / Methods | | AlphaFLOW-MD | | MSA subsampling | | | | MDGen | Ours | | |
| --- | --- | --- | --- | --- | --- | --- | --- | --- | --- | --- | --- |
| | | Full | Distilled | 32 | 48 | 64 | 256 | | Pre-train | EBA-DPO | EBA |
| Predicting flexibility | Pairwise RMSD $r \uparrow$ | 0.48 | 0.48 | 0.03 | 0.12 | 0.22 | 0.15 | 0.48 | 0.43 | 0.59 | **0.62** |
| | Global RMSF $r \uparrow$ | 0.60 | 0.54 | 0.13 | 0.23 | 0.29 | 0.26 | 0.50 | 0.50 | 0.69 | **0.71** |
| | Per-target RMSF $r \uparrow$ | 0.85 | 0.81 | 0.51 | 0.52 | 0.51 | 0.55 | 0.71 | 0.72 | **0.90** | **0.90** |
| Distributional accuracy | Root mean $\mathcal{W}_2$-dist. $\downarrow$ | 2.61 | 3.70 | 6.15 | 5.32 | 4.28 | 3.62 | 2.69 | 3.22 | **2.43** | **2.43** |
| | $\hookrightarrow$ Trans. contrib. $\downarrow$ | 2.28 | 3.10 | 5.22 | 3.92 | 3.33 | 2.87 | - | 2.47 | 2.05 | 2.03 |
| | $\hookrightarrow$ Var. contrib. $\downarrow$ | 1.30 | 1.52 | 3.55 | 2.49 | 2.24 | 2.24 | - | 1.89 | **1.20** | 1.20 |
| | MD PCA $\mathcal{W}_2$-dist. $\downarrow$ | 1.52 | 1.73 | 2.44 | 2.30 | 2.23 | 1.88 | 1.89 | 1.78 | 1.20 | **1.19** |
| | Joint PCA $\mathcal{W}_2$-dist. $\downarrow$ | 2.25 | 3.05 | 5.51 | 4.51 | 3.57 | 3.02 | - | 2.47 | 2.08 | **2.04** |
| | % PC-sim $> 0.5 \uparrow$ | **44** | 34 | 15 | 18 | 21 | 21 | - | 28 | 38 | **44** |
| Ensemble observables | Weak contacts $J \uparrow$ | 0.62 | 0.52 | 0.40 | 0.40 | 0.37 | 0.30 | 0.51 | 0.23 | 0.63 | **0.65** |
| | Transient contacts $J \uparrow$ | **0.41** | 0.28 | 0.23 | 0.26 | 0.27 | 0.27 | - | 0.25 | 0.38 | 0.41 |
| | Exposed residue $J \uparrow$ | 0.50 | 0.48 | 0.34 | 0.37 | 0.37 | 0.33 | 0.29 | 0.29 | 0.68 | **0.70** |
| | Exposed MI matrix $\rho \uparrow$ | 0.25 | 0.14 | 0.14 | 0.11 | 0.10 | 0.06 | - | 0.01 | 0.35 | **0.36** |
| Runtime | GPU sec. per sample | 70 | 7 | | 4 | | | 0.2 | | 0.9 | |

Specifically, for each structure, we apply the Smooth LDDT loss (Abramson et al., 2024) which is defined as:

$$L_{\text{Smooth LDDT}} =$$
$$\frac{1}{4} \frac{\text{mean}_{l \neq m} \sum_{k=1}^{4} \sigma(\delta_k - \Delta D_{i,j}) \mathbf{1}\{D_{:,i,j}^{\text{gt}} < 15 \text{ Å}\}}{\text{mean}_{l \neq m} \mathbf{1}\{D_{:,i,j}^{\text{gt}} < 15 \text{ Å}\}}, \quad (13)$$

where $\Delta D_{i,j} \triangleq \text{abs}(D_{i,j}^{\text{pred}} - D_{:,i,j}^{\text{gt}}) \in \mathbb{R}^{N \times N}$, $D^{\text{pred}} = \text{Dist}(\hat{\mathbf{x}}_0) \in \mathbb{R}^{N \times N}$ and $D^{\text{gt}} = \text{Dist}(\mathbf{x}_0) \in \mathbb{R}^{N \times N}$ represent the pairwise distance matrices of predicted and ground truth structures, and $\sigma(\cdot)$ indicates the sigmoid function. This term penalizes deviations in pairwise distances, encouraging the preservation of local and global structural features.

The final denoising training loss is a weighted combination of the two components above:

$$L_{\text{total}} = \lambda_{\text{mse}} L_{\text{Aligned MSE}} + \lambda_{\text{lddt}} L_{\text{Smooth LDDT}}. \quad (14)$$

$\lambda_{\text{mse}}, \lambda_{\text{lddt}} \geq 0$ are hyperparameters that control the relative importance of the aligned MSE and smooth LDDT terms. Putting it together, we compute the MSE term in Eq. (10) by Eq. (14), which ensures both structural alignment and geometric accuracy and enables the model to better capture biologically relevant conformations. EBA fine-tuning procedure for diffusion models is detailed in Algorithm 1.

## 4. Experiments

### 4.1. Setup

To demonstrate the effectiveness of the proposed fine-tuning pipeline, we evaluate the protein ensemble generation task on the ATLAS dataset (Vander Meersche et al., 2024) following the benchmark in Jing et al. (2024a). For baselines, we mainly compare our methods against: AlphaFold2 (Jumper et al., 2021) with MSA subsampling (Del Alamo et al., 2022), AlphaFlow (Jing et al., 2024a) and MDGen (Jing et al., 2024b). We follow the evaluation metrics introduced in Jing et al. (2024a) and classify them as below:

- Flexibility correlation ($\uparrow$): the pearson correlation $r$ of the pairwise RMSD, global root-mean-square-fluctutation (RMSF) and per-target RMSF.

- Distributional accuracy: Root mean of 2-Wasserstein distance ($\mathcal{W}_2$-dist) and its translation and variance contribution ($\downarrow$), MD PCA $\mathcal{W}_2$-dist ($\downarrow$), joint PCA $\mathcal{W}_2$-dist ($\downarrow$); the percentage of samples with PC-sim $> 0.5$ ($\uparrow$).

- Ensemble observables ($\uparrow$): the Jaccard similarity $\mathcal{J}$ of the weak contacts, transient contacts, and exposed residue as well as the Spearman correlation $\rho$ of the exposed mutual information (MI) matrix.

### 4.2. Training Pipeline

We adopt Protenix (Chen et al., 2025) as AlphaFold3 architecture implementation and initialize the model with the pre-trained parameters from the released weights. No template input is used across our study. The ATLAS trajectories are pre-processed by parsing, following Jing et al. (2024a), and by applying the training data pipeline in the open-sourced Protenix codebase. We fine-tune the pre-trained DIFFUSION MODULE in two stages including supervised fine-tuning and physical alignment, with the parameters of other modules frozen.

**Stage 1: Supervised Fine-tuning.** In the first stage, the pre-trained model is adapted via fine-tuning to the ATLAS simulated trajectories (Vander Meersche et al., 2024). AT-LAS provides diverse ensembles of protein conformations generated via all-atom MD simulations. During this phase, the model is trained to minimize the vanilla diffusion loss. This fine-tuning stage enables the diffusion to coarsely adapt to the data distribution over conformational space sampled by the MD simulators across different targets.

**Stage 2: Physical Alignment.** The second stage employs the EBA alignment to align the diffusion with physical energy feedback. We use (1) Direct Preference Optimization (EBA-DPO) that leverages energy-agnostic preference pairs (binary win-lose pairs) to align the model's predictions, emphasizing conformational states with favorable energy profiles; (2) the proposed EBA objective to account for the energy difference $\Delta E$ between multiple conformational states weighted by physical energies. To make training more efficient, we collect the ATLAS training set and annotate their potential energy *off-policy* with local minimization. The detailed protocol can be found in Appendix C.

In practice, we also observed that $E(\mathbf{x})$ varies significantly with protein size (the number of residues or atoms), making the energy-informed training unstable due to large variance in the objective. Inspired by Naganathan & Muñoz (2005), which shows that *the folding time scales with the number of residues* (with an exponent of 0.5), we introduce a sample-specific regularization factor $L^{0.5}$ to normalize the energy. Concretely, we rescale by multiplying $\beta$ with $1/L^{0.5}$ for each sample, where $L$ denotes the number of residues in $\mathbf{x}$.

### 4.3. Benchmark on Molecular Dynamics Ensembles

We evaluate the models to emulate the conformation ensembles of ATLAS MD simulation dataset (Vander Meersche et al., 2024), which includes three replicas of 100ns production trajectories of 1390 protein targets in total. We strictly follow the experimental settings as well as the data split in Jing et al. (2024a) and sample 250 predictions per test target using different models. Baseline evaluations are taken from the original tabular results in Jing et al. (2024a) and Jing et al. (2024b). As shown in Table 1, our methods consistently outperform baseline models as well as seeing an improvement than EBA-DPO (an implementation of DPO within our framework), where we set $K = 2$ for EBA model to make fair comparison with DPO. The EBA model shows significant improvement in exposed residue $\mathcal{J}$ and the exposed MI matrix $\rho$, along with high RMSF correlation $r$. This suggests that our model captures collective long-range dynamics by exposing buried residues to solvent, as a physically plausible behavior where other baselines perform poorly. As an illustration, we visualize the sampled ensembles for the test target 6uof_A following Jing et al. (2024a)

and plot the C$\alpha$-RMSF against the residue index in Figure 3.

*Table 2.* Ablation results on MD ensemble benchmark of ATLAS test set (Jing et al., 2024a) for EBA with different number of candidate samples K in mini-batch during alignment training.

| Metrics | $K=2$ | $K=3$ | $K=5$ |
|---|---|---|---|
| Pairwise RMSD $r \uparrow$ | 0.62 | 0.61 | 0.62 |
| Global RMSF $r \uparrow$ | 0.71 | 0.71 | 0.72 |
| Per-target RMSF $r \uparrow$ | 0.90 | 0.90 | 0.89 |
| Root mean $\mathcal{W}_2$-dist. $\downarrow$ | 2.43 | 2.42 | 2.40 |
| $\hookrightarrow$ Trans. contrib. $\downarrow$ | 2.03 | 2.02 | 2.05 |
| $\hookrightarrow$ Var. contrib. $\downarrow$ | 1.20 | 1.18 | 1.25 |
| MD PCA $\mathcal{W}_2$-dist. $\downarrow$ | 1.19 | 1.18 | 1.16 |
| Joint PCA $\mathcal{W}_2$-dist. $\downarrow$ | 2.04 | 2.04 | 2.15 |
| % PC-sim $> 0.5 \uparrow$ | 44 | 39 | 43 |
| Weak contacts $\mathcal{J} \uparrow$ | 0.65 | 0.65 | 0.65 |
| Transient contacts $\mathcal{J} \uparrow$ | 0.41 | 0.41 | 0.39 |
| Exposed residue $\mathcal{J} \uparrow$ | 0.70 | 0.69 | 0.70 |
| Exposed MI matrix $\rho \uparrow$ | 0.36 | 0.37 | 0.34 |
| Iteration per step (s) | 4.3 | 5.4 | 7.8 |
| Avg. GPU memory (GB) | 12.0 | 13.9 | 16.3 |

### 4.4. Ablation Study

To assess the impact of the size $K$ of mini-batch on the performance of EBA during fine-tuning, we further conduct an ablation study on the MD ensemble benchmark of the ATLAS test set. Table 2 presents the results for varying values of $K$, demonstrating the robustness of our mini-batch EBA objective Eq. (10). Note that large $K$ is at the cost of increased computational overhead, as the model's forward pass scales linearly with $K$. Also, we investigate the effect of $\lambda$ and $\eta$, *scaling factors* for noise and *backward step size* respectively, on the quality of generated ensemble in Fig. 4.

Additionally, we conduct a simple profiling on different $K(= 2, 3, 5)$ regarding the training speed and GPU memory utilization. The DDP parallelism and $4\times$ A100 40GB GPUs are used for this benchmarking. As shown in Table 2, the increase of $K$ does not introduce heavy memory overhead, while the iteration step grows in a quasi-linear trend.

### 4.5. Discussion

Our approach outperform existing methods in MD ensemble generation based off AlphaFold3's direct atomic interaction modeling, which reduces the need for coarse-graining or internal coordinates while allowing for more precise capture of fine-grained conformational changes. Beyond data-driven training, the model is fine-tuned with physical feedback, ensuring that learned structures are not only statistically plausible but also physically consistent. Especially, through EBA, we align the model to (approximately) emulate the Boltzmann distribution, enhancing its capability to generate

thermodynamics-consistent ensembles. Our adaptation effectively bridges neural and physical methods as a promising framework for protein conformation ensemble generation.

Importantly, although the derivation sums over only $K$ mini-batch samples, EBA is not designed to approximate the intractable partition function $Z$ in Eq. 3 (which is beyond the scope of this paper), since doing so introduces non-negligible truncation error. Rather, the EBA objective leverages the Boltzmann factors, i.e., the relative weights among conformations within each sampled mini-batch, which remain invariant to the batch size and adhere to partial knowledge derived from the Boltzmann distribution.

## 5. Related work

**Protein conformation generation.** Unlike structure prediction (Jumper et al., 2021) aiming to identify a single, most-likely folded structure, protein conformation generation focuses on sampling an ensemble of physically plausible states that capture the underlying energy landscape. Boltzmann generator (Noé et al., 2019) leverages normalizing flows to approximate the Boltzmann distribution by training on simulation data. Arts et al. (2023) applies the diffusion model to capture such distribution over coarse-grained protein conformations. EigenFold (Jing et al., 2023) adopts a generative perspective on structure prediction, enabling the generation of multiple structures given an input sequence. Str2Str (Lu et al., 2024b) introduces a score-based sampler trained exclusively on PDB data, framing conformation generation in a structure-to-structure paradigm. DiG (Zheng et al., 2024) trains a conditional diffusion model on both PDB and in-house simulation data. ConfDiff (Wang et al., 2024) incorporates the energy- and force-guidance during the reverse process of diffusion to enhance the accuracy of conformation generation. AlphaFlow (Jing et al., 2024a) repurposes the AlphaFold2 model into a denoising network via flow matching. ESMDiff (Lu et al., 2024a) fine-tunes the protein language model ESM3 using discrete diffusion to produce protein conformations. Finally, MDGen (Jing et al., 2024b) attempts direct generation of MD trajectories by modeling them as time-series of protein structures.

**Alignment methods for generative models.** Aligning generative models with desired objectives is becoming increasingly important. The Reinforcement Learning from Human Feedback (RLHF) framework optimizes models via RL using human preference rewards and has been widely applied in tasks like machine translation (Kreutzer et al., 2018), summarization (Stiennon et al., 2020), and instruction following (Ouyang et al., 2022). RLHF has also been applied for alignment of text-to-image diffusion models (Black et al., 2023; Fan et al., 2024). However, RL-based fine-tuning faces significant challenges in stability and scalability. Direct Preference Optimization (Rafailov et al., 2024) mit-

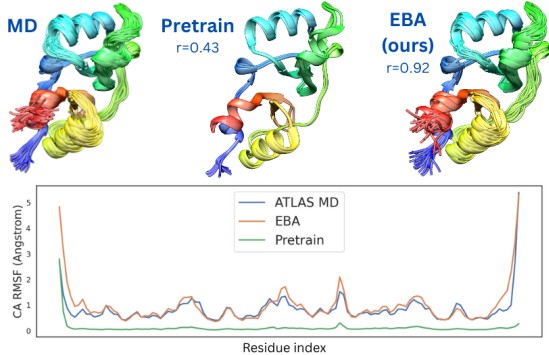

*Figure 3.* (Top) Structure ensembles for the target `6uof_A` in ATLAS test set with RMSF correlation $r$ labeled. (Bottom) C$\alpha$-RMSF versus the residue index (N $\rightarrow$ C terminus from left to right).

igates these issues by directly optimizing for the optimal policy via re-parameterization of an implicit reward model. This approach has been extended beyond language modeling: Diffusion-DPO (Wallace et al., 2024) for text-to-image generation, ABDPO (Zhou et al., 2024) for antibody design using Rosetta energy (Alford et al., 2017), and ALIDIFF (Gu et al., 2024) and DECOMPDPO (Cheng et al., 2024) for molecular optimization in structure-based drug design.

*Remarks: Our method differs from existing approaches above by adopting a more general-form objective, being grounded in physically meaningful motivations, addressing a different task and demonstrating superior performance.*

## 6. Conclusion and Limitations

**Conclusion.** In this study, we introduced a physics-inspired alignment framework, namely EBA, for protein ensemble generation. EBA leverages energy feedback to refine the pre-trained diffusion models based on the Boltzmann factor between different conformational states. Our approach effectively bridges structure data with physical signal, enabling scalable and physically grounded model alignment. Built upon AlphaFold3, the fine-tuned diffusion model demonstrates the effectiveness of the proposed EBA by benchmarking on the ATLAS MD dataset. Results underscore the potential of incorporating physical energy supervision into the data-drive models, advancing more accurate and thermodynamically consistent ensemble generation. Our method opens new avenues for applications in modern drug discovery and biomolecular simulations.

**Limitations.** Despite its advantages, our approach has several limitations. First, since the base model, AlphaFold3, is originally designed for folding, its fine-tuned variant may not be readily well-suited for modeling long-timescale dynamics (Lindorff-Larsen et al., 2011). Second, the accuracy of the employed energy (force fields) needs further

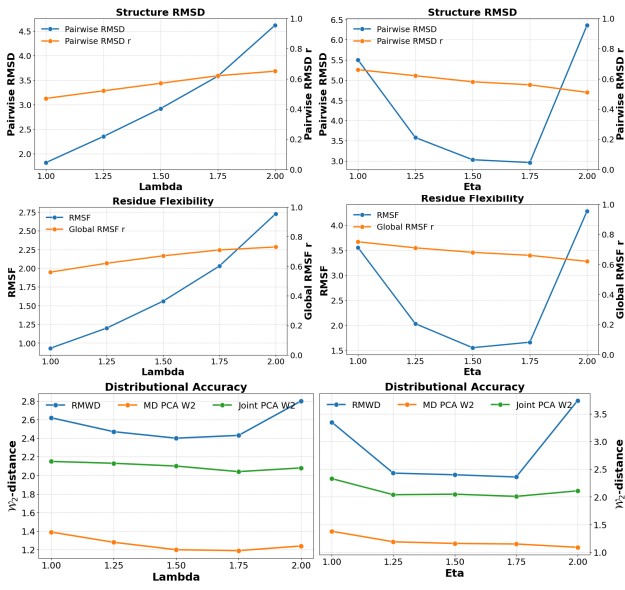

*Figure 4.* Evaluation performances versus different values of inference hyperparameters including *scaling factors* for noise $\lambda$ and and *backward step size* $\eta$ during diffusion sampling ($N = 250$).

improvement, as their precision falls short of quantum-level single-point energy calculations. Third, our current study is restricted to generating single-chain protein ensemble. Finally, we have solely implemented and evaluated EBA within the diffusion framework, leaving open the exploration of alternative generative models. Future work will focus on addressing these limitations by extending to broader biomolecular systems, incorporating quantum-level energy calculation, and exploring more alignment implementations.

## Acknowledgements

We thank Meihua Dang, Qijia He, Dinghuai Zhang, Zuobai Zhang as well as anonymous ICML reviewers for their helpful feedback and discussions. We also acknowledge the authors of Protenix for open-sourcing the code of training pipeline and model implementation. The authors acknowledge funding from the Canada CIFAR AI Chair Program and the Intel-Mila partnership program. The computation resource of this project is supported by Mila and the Digital Research Alliance of Canada.

## Impact Statement

This study advances the intersection of generative modeling and computational biology, enhancing protein structure prediction and conformation sampling with potential applications in drug discovery and biotechnology. While these advancements hold promise for scientific innovation, we acknowledge potential risks associated with AI-driven biological molecular design, including the misuse of generative models for unintended purposes. Responsible use and oversight in applying these models to sensitive biological domains are important considerations. However, we do not elaborate further on specific ethical concerns, as the studied problem and related techniques are well established in the field of machine learning.

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

# A. Theoretical Results

## A.1. Proof of Equation 9

For simplicity, we denote the decomposed negative KL-divergence term $-\mathbb{D}_{\mathrm{KL}}[q(\mathbf{x}_{t-1}^j|\mathbf{x}_{0,t}^j)\|p_\theta(\mathbf{x}_{t-1}^j|\mathbf{x}_t^j,\mathbf{c})]$ as $F(x^j,t)$, where $t$ is the index of time step and $j$ is the sample index. Let $w(\mathbf{x}_i) \triangleq \frac{e^{-\beta E(\mathbf{x}^i;\mathbf{c})}}{\sum_j^K e^{-\beta E(\mathbf{x}^j;\mathbf{c})}}(i=1,\dots,K)$ be the Boltzmann weight. We aim to establish that:

$$-\mathbb{E}_{(c,\{\mathbf{x}\}_1^K)\sim D}\sum_{i=1}^K w(\mathbf{x}_i)\log\frac{e^{\sum_t \mathbb{E}_{\mathbf{x}_t^i\sim q(\mathbf{x}_t^i|\mathbf{x}_0^i)}F(x^i,t)}}{\sum_j^K e^{\sum_t \mathbb{E}_{\mathbf{x}_t^j\sim q(\mathbf{x}_t^j|\mathbf{x}_0^j)}F(x^j,t)}}$$

$$\leq -\mathbb{E}_{(c,\{\mathbf{x}\}_1^K)\sim D}\sum_{t=1}^T\sum_{i=1}^K \mathbb{E}_{\mathbf{x}_t^i\sim q(\mathbf{x}_t^i|\mathbf{x}_0^i),\forall i}\left[w(\mathbf{x}_i)\log\frac{e^{F(x^i,t)}}{\sum_j^K e^{F(x^j,t)}}\right]. \tag{15}$$

Firstly, we rewrite the left-hand side by expanding the log:

The left-hand side is:

$$-\mathbb{E}_{(c,\{\mathbf{x}\}_1^K)\sim D}\sum_{i=1}^K w(\mathbf{x}_i)\log\frac{e^{\sum_t \mathbb{E}_{\mathbf{x}_t^i\sim q(\mathbf{x}_t^i|\mathbf{x}_0^i)}F(x^i,t)}}{\sum_j^K e^{\sum_t \mathbb{E}_{\mathbf{x}_t^j\sim q(\mathbf{x}_t^j|\mathbf{x}_0^j)}F(x^j,t)}}$$

$$= -\mathbb{E}_{(c,\{\mathbf{x}\}_1^K)\sim D}\sum_{i=1}^K w(\mathbf{x}_i)\left[\sum_t \mathbb{E}_{\mathbf{x}_t^i\sim q(\mathbf{x}_t^i|\mathbf{x}_0^i)}F(x^i,t)-\log\sum_{j=1}^K e^{\sum_t \mathbb{E}_{\mathbf{x}_t^j\sim q(\mathbf{x}_t^j|\mathbf{x}_0^j)}F(x^j,t)}\right]$$

$$= \mathbb{E}_{(c,\{\mathbf{x}\}_1^K)\sim D}\sum_{i=1}^K w(\mathbf{x}_i)\left[-\sum_{t=1}^T \mathbb{E}_{\mathbf{x}_t^i\sim q(\mathbf{x}_t^i|\mathbf{x}_0^i)}F(x^i,t)+\log\sum_{j=1}^K e^{\sum_t \mathbb{E}_{\mathbf{x}_t^j\sim q(\mathbf{x}_t^j|\mathbf{x}_0^j)}F(x^j,t)}\right]$$

For the denominator term:

$$\log\sum_{j=1}^K e^{\sum_t \mathbb{E}_{\mathbf{x}_t^j\sim q(\mathbf{x}_t^j|\mathbf{x}_0^j)}F(x^j,t)},$$

we notice that this is the log-sum-exp (LSE) function. Apply the *Jensen's inequality* for with convexity of the LSE function and we can pull the summation and expectation outside:

$$\log\sum_{j=1}^K e^{\sum_t \mathbb{E}_{\mathbf{x}_t^j\sim q(\mathbf{x}_t^j|\mathbf{x}_0^j)}[F(x^j,t)]}\leq\sum_{t=1}^T \mathbb{E}_{\mathbf{x}_t^j\sim q(\mathbf{x}_t^j|\mathbf{x}_0^j),\forall j}\log\sum_{j=1}^K e^{F(x^j,t)}.$$

Thus, the left-hand side satisfies:

$$-\log\frac{e^{\sum_t \mathbb{E}_{\mathbf{x}_t^j\sim q(\mathbf{x}_t^j|\mathbf{x}_0^j)}[F(x^i,t)]}}{\sum_j e^{\sum_t \mathbb{E}_{\mathbf{x}_t^j\sim q(\mathbf{x}_t^j|\mathbf{x}_0^j)}[F(x^j,t)]}}\leq-\sum_{t=1}^T \mathbb{E}_{\mathbf{x}_t^j\sim q(\mathbf{x}_t^j|\mathbf{x}_0^j),\forall j}\log\frac{e^{F(x^i,t)}}{\sum_j e^{F(x^j,t)}}.$$

Because the Boltzmann weights are strictly positive, $\sum_{i=1}^K w(\mathbf{x}_i)$ is a convex linear combination, following the inequality above and thus we have:

$$-\mathbb{E}_{(c,\{\mathbf{x}\}_1^K)\sim D}\sum_{i=1}^K w(\mathbf{x}_i)\log\frac{e^{\sum_t \mathbb{E}_{\mathbf{x}_t^i\sim q(\mathbf{x}_t^i|\mathbf{x}_0^i)}F(x^i,t)}}{\sum_j^K e^{\sum_t \mathbb{E}_{\mathbf{x}_t^j\sim q(\mathbf{x}_t^j|\mathbf{x}_0^j)},F(x^j,t)}}\leq-\mathbb{E}_{(c,\{\mathbf{x}\}_1^K)\sim D}\sum_{t=1}^T\sum_{i=1}^K \mathbb{E}_{\mathbf{x}_t^i\sim q(\mathbf{x}_t^i|\mathbf{x}_0^i),\forall i}\left[w(\mathbf{x}_i)\log\frac{e^{F(x^i,t)}}{\sum_j^K e^{F(x^j,t)}}\right].$$

The inequality is now proved. $\square$

### A.2. Derivation for EBA-DPO-Diffusion

To address the intractability of $p_\theta(\mathbf{x}|\mathbf{c})$ for diffusion models, Wallace et al. (2024) has derived an ELBO objective for diffusion DPO. We provide a brief summary of their derivation here and refer the readers to Wallace et al. (2024) for the complete details.

The diffusion ELBO for the objective in Eq. 11 is:

$$\mathcal{L}_{\text{EBA-DPO-Diffusion}}(\theta) \leq -\mathbb{E}_{(c,\{\mathbf{x}\}_1^K)\sim D, t\sim\mathcal{U}(0,T), \mathbf{x}_t^i\sim p_\theta(\mathbf{x}_t^i|\mathbf{x}_0^i, \mathbf{c})} \log\sigma\left(\alpha\log\frac{p_\theta(\mathbf{x}_{t-1}^1|\mathbf{x}_t^1, \mathbf{c})}{p_{\text{ref}}(\mathbf{x}_{t-1}^1|\mathbf{x}_t^1, \mathbf{c})} - \alpha\log\frac{p_\theta(\mathbf{x}_{t-1}^2|\mathbf{x}_t^2, \mathbf{c})}{p_{\text{ref}}(\mathbf{x}_{t-1}^2|\mathbf{x}_t^2, \mathbf{c})}\right). \tag{16}$$

The reverse process $p_\theta(\mathbf{x}_{1:T}|\mathbf{x}_0)$ can be further approximated by the forward process $q(\mathbf{x}_{1:T}|\mathbf{x}_0)$:

$$\mathcal{L}_{\text{EBA-DPO-Diffusion}}(\theta) = -\mathbb{E}_{(c,\{\mathbf{x}\}_1^K)\sim D, t\sim\mathcal{U}(0,T), \mathbf{x}_t^i\sim q(\mathbf{x}_t^i|\mathbf{x}_0^i)}\log\sigma(-\alpha T(\\
\mathbb{D}_{\text{KL}}[q(\mathbf{x}_{t-1}^1|\mathbf{x}_t^1, \mathbf{x}_0^i)\|p_\theta(\mathbf{x}_{t-1}^1|\mathbf{x}_t^1, \mathbf{c})] - \mathbb{D}_{\text{KL}}[q(\mathbf{x}_{t-1}^1|\mathbf{x}_t^1, \mathbf{x}_0^i)\|p_{\text{ref}}(\mathbf{x}_{t-1}^1|\mathbf{x}_t^1, \mathbf{c})]\\
- \mathbb{D}_{\text{KL}}[q(\mathbf{x}_{t-1}^2|\mathbf{x}_t^2, \mathbf{x}_0^i)\|p_\theta(\mathbf{x}_{t-1}^2|\mathbf{x}_t^2, \mathbf{c})] + \mathbb{D}_{\text{KL}}[q(\mathbf{x}_{t-1}^2|\mathbf{x}_t^2, \mathbf{x}_0^i)\|p_{\text{ref}}(\mathbf{x}_{t-1}^2|\mathbf{x}_t^2, \mathbf{c})])). \tag{17}$$

With some algebra, the loss simplifies to:

$$\mathcal{L}_{\text{EBA-DPO-Diffusion}}(\theta) = -\mathbb{E}_{(c,\{\mathbf{x}\}_1^K)\sim D, t\sim\mathcal{U}(0,T), \mathbf{x}_t^i\sim q(\mathbf{x}_t^i|\mathbf{x}_0^i)}\log\sigma(-\alpha T(\\
\|\epsilon^1 - \epsilon_\theta(\mathbf{x}_t^1, t, \mathbf{c})\|_2^2 - \|\epsilon^1 - \epsilon_{\text{ref}}(\mathbf{x}_t^1, t, \mathbf{c})\|_2^2 - \|\epsilon^2 - \epsilon_\theta(\mathbf{x}_t^2, t, \mathbf{c})\|_2^2 + \|\epsilon^2 - \epsilon_{\text{ref}}(\mathbf{x}_t^2, t, \mathbf{c})\|_2^2)). \tag{18}$$

In practice, we combine $\alpha$ and $T$ into a single hyperparameter, denoted as $T$. Similar to EBA-Diffusion in Algorithm 1, the MSE terms are computed by Eq. 14.

### A.3. Further Discussion on the Learning Objective

Here we show the optimization of $\mathbb{D}_{\text{KL}}(p_B^K\|p_\theta^K)$ regarding parameters $\theta$ in Eq. 6 will finally converge to a minimizer of $\mathbb{D}_{\text{KL}}(p_B\|p_\theta)$ in the following sense. Suppose the $\theta$ is a global minimizer of $\mathbb{D}_{\text{KL}}(p_B^K\|p_\theta^K)$ w.r.t. $\forall\{x^j\}_{j=1}^K\sim p_B$. Let $K=2, \forall x^i, x^j$, we have $p_B^K = p_\theta^K$, which implies that $(a, b > 0)$:

$$\exp(-bE(x^i))/\exp(-bE(x^j)) = \exp(-aE_\theta(x^i))/\exp(-aE_\theta(x^j)) \tag{19}$$

Then we see $-b[E(x^i) - E(x^j)] = -a[E_\theta(x^i) - E_\theta(x^j)]$, or equivalently

$$E_\theta(x^i) = \frac{b}{a}E(x^i) + [E_\theta(x^j) - \frac{b}{a}E(x^j)], \forall i, j$$

Note that by marginalizing $j$, we see that

$$E_\theta(x^i) = \frac{b}{a}E(x^i) + Const, \forall i$$

Finally we plug in, $\forall x^i$,

$$p_\theta(x^i) = \exp(-aE_\theta(x^i))/Z = \exp(-a(\frac{b}{a}E(x^i) + Const))/Z = \exp(-bE(x^i))/Z' = p_B(x^i),$$

or $p_\theta(x) = p_B(x)$. This indicates that model alignment according to Boltzmann factor can also provide useful guidance towards the underlying Boltzmann distribution: when the equality Eq. 19 holds for any pair of $x^i, x^j$, then we immediately have $p_\theta \equiv p_B$.

# B. Additional Experimental Results

In this section, we additionally incorporate more reports as comparison to give interested readers better overview of our methods as shown in Table S1. Results of ConfDiff (Wang et al., 2024) and BioEmu (Lewis et al., 2024) are obtained by performing inference pipeline using code and checkpoints from their official repositories. For the ConfDiff, we use the checkpoint named `OpenFold-r3-MD` which was finetuned on the ATLAS dataset (Wang et al., 2024). Notably, the proposed EBA consistently outperformed the pre-trained and the Stage2-SFT checkpoints (after two stages of supervised fine-tuning, see Section 4.2), which demonstrates the proposed EBA objective provides additional knowledge from the energy landscape beyond the maximizing likelihood training.

*Table S1.* Supplementary comparison on MD ensemble benchmark of ATLAS test set following Jing et al. (2024a) where the median across all test targets is reported.

| Metrics / Methods | | ESMFLOW-MD | | | | | Ours | | |
|---|---|---|---|---|---|---|---|---|---|
| | | Full | Distilled | AlphaFold | ConfDiff | BioEmu | Pre-train | Stage2-SFT | EBA |
| Predicting flexibility | Pairwise RMSD $r \uparrow$ | 0.19 | 0.19 | 0.10 | 0.59 | 0.46 | 0.43 | 0.57 | **0.62** |
| | Global RMSF $r \uparrow$ | 0.31 | 0.33 | 0.21 | 0.67 | 0.57 | 0.50 | 0.69 | **0.71** |
| | Per-target RMSF $r \uparrow$ | 0.76 | 0.74 | 0.52 | 0.85 | 0.71 | 0.72 | 0.89 | **0.90** |
| Distributional accuracy | Root mean $\mathcal{W}_2$-dist. $\downarrow$ | 3.60 | 4.23 | 3.58 | 2.76 | 4.32 | 3.22 | 2.58 | **2.43** |
| | $\hookrightarrow$ Trans. contrib. $\downarrow$ | 3.13 | 3.75 | 2.86 | 2.23 | 4.04 | 2.47 | 2.15 | **2.03** |
| | $\hookrightarrow$ Var. contrib. $\downarrow$ | 1.74 | 1.90 | 2.27 | 1.40 | 1.77 | 1.89 | 1.28 | **1.20** |
| | MD PCA $\mathcal{W}_2$-dist. $\downarrow$ | 1.51 | 1.87 | 1.99 | 1.44 | 1.97 | 1.78 | 1.29 | **1.19** |
| | Joint PCA $\mathcal{W}_2$-dist. $\downarrow$ | 3.19 | 3.79 | 1.58 | 2.25 | 3.98 | 2.47 | 2.13 | **2.04** |
| | % PC-sim $> 0.5 \uparrow$ | 26 | 33 | 23 | 35 | **51** | 28 | 42 | 44 |
| Ensemble observables | Weak contacts $J \uparrow$ | 0.55 | 0.48 | 0.23 | 0.59 | 0.33 | 0.23 | 0.63 | **0.65** |
| | Transient contacts $J \uparrow$ | 0.34 | 0.30 | 0.28 | 0.36 | - | 0.25 | **0.43** | 0.41 |
| | Exposed residue $J \uparrow$ | 0.49 | 0.43 | 0.32 | 0.50 | - | 0.29 | 0.67 | **0.70** |
| | Exposed MI matrix $\rho \uparrow$ | 0.20 | 0.16 | 0.02 | 0.24 | 0.07 | 0.01 | 0.35 | **0.36** |

# C. Implementation Details

**Rigid align.** Algorithm 3 describes the rigid alignment procedure used for atomic protein structures during loss calculation, which follows the Kabsch algorithm with reflection consideration. Notably, the structural alignment is in practice performed by transforming the ground truth structure to match the predicted structure, allowing gradients to propagate correctly during optimization, *i.e.*, $\mathbf{x}^{\text{ref}} \leftarrow \hat{\mathbf{x}}$.

**Diffusion inference.** During inference, Algorithm 2 outlines the diffusion module sampling procedure, which iteratively refines an initial conformation drawn from a Gaussian distribution. At each step, a randomly sample rigid transformation is applied, followed by the standard denoising step using a learned diffusion module. In Algorithm 2, $\mathcal{N}$ is the Gaussian distribution; the $\mathcal{U}_{SO(3)}$ indicates the uniformly distribution rotations in three dimensions and is implemented using the `scipy.spatial.transform.Rotation.random`. We use $\gamma_0 = 0.8, \gamma_{min} = 1.0, N_{step} = 20, \lambda = 1.75, \eta = 1.25$ as the default hyperparameters for sample diffusion across all finetuned and aligned models; for pre-trained model, we keep the default $\lambda = 1.5, \eta = 1.5$ since we find that the change of scaling factors will yield significantly worse results for diffusion module without fine-tuning. The inference noise scheduler also has the same configuration as Abramson et al. (2024), i.e., $s_{max} = 160.0, s_{min} = 4 \times 10^{-4}, p = 7$ and $\sigma_{data} = 16$.

**Training data.** For training, we follow the dataset splitting of Jing et al. (2024a) for ATLAS. Specifically, we download the ATLAS MD trajectories, which comprises 1,390 proteins selected for structural diversity based on ECOD domain classification. This results in train / validation / test splits of 1,266 / 39 / 82 MD ensembles, with the rest excluded due to excessive sequence length (Jing et al., 2024a) . The conformation are stratified sampled from the production trajectories with an interval as $1ns$ (per 100 frames in the original saving frequency). We combine the three replicas as the final ensemble with 300 structures per target. The multiple sequence alignments (MSA) for each target are downloaded from the pre-computed deposit in Protenix training data (Chen et al., 2018). For the energy-based alignment fine-tuning, we employ an off-policy strategy where energy labels are assigned by scanning the entire training dataset using a force field prior to loading the data for model training. This allows us to efficiently decouple data processing from model updates to accelerate training.

---

**Algorithm 2** Inference of Diffusion Module (Algo. 18 in Abramson et al. (2024))

---

**Require:** Features $f^*$; embeddings $s_i^{\text{inputs}}, s_i^{\text{emb}}, z_{ij}^{\text{emb}}$; noise schedule $[c_0, c_1, \ldots, c_T]$
**Require:** Hyperparameter: $\gamma_0 = 0.8, \gamma_{\min} = 1.0$, noise scale $\lambda = 1.5$, step scale $\eta = 1.25$
1: $\mathbf{x}_l \sim c_0 \cdot \mathcal{N}(\mathbf{0}, I_3)$ // Initialization
2: **for** $c_\tau \in c_1, \ldots, c_T$ **do**
3:      $\mathbf{x}_l \leftarrow \mathbf{x}_l - \text{mean}_l \mathbf{x}_l$ // Center coordinates
4:      $R \sim \mathcal{U}_{SO(3)}, \boldsymbol{t} \sim \mathcal{N}(\mathbf{0}, I_3)$
5:      $\mathbf{x}_l \leftarrow R \cdot \mathbf{x}_l + \boldsymbol{t}, \forall l$ // Apply roto-translate
6:      $\gamma \leftarrow \gamma_0$ if $c_\tau > \gamma_{\min}$ else 0
7:      $\hat{t} \leftarrow c_{\tau-1}(\gamma + 1)$
8:      $\epsilon_l \leftarrow \lambda \sqrt{\hat{t}^2 - c_{\tau-1}^2} \cdot \mathcal{N}(\mathbf{0}, I_3)$
9:      $\mathbf{x}_l^{\text{noisy}} \leftarrow \mathbf{x}_l + \epsilon_l$
10:      $\mathbf{x}_l^{\text{denoised}} \leftarrow \text{DiffusionModule}(\mathbf{x}_l^{\text{noisy}}, \hat{t}, s_i^{\text{inputs}}, s_i^{\text{emb}}, z_{ij}^{\text{emb}}, f^*)$
11:      $\delta_l \leftarrow (\mathbf{x}_l^{\text{noisy}} - \mathbf{x}_l^{\text{denoised}})/\hat{t}$
12:      $dt \leftarrow c_\tau - \hat{t}$
13:      $\mathbf{x}_l \leftarrow \mathbf{x}_l^{\text{noisy}} + \eta \cdot dt \cdot \delta_l$
14: **end for**
15: **Return** $\mathbf{x}_l$

---

**Algorithm 3** Structure Rigid Align (Kabsch-Umeyama Algorithm)

---

**Require:** $\mathbf{x}_l, \mathbf{x}_l^{\text{ref}}$
1: $\mu, \mu^{\text{ref}} \leftarrow \frac{1}{N} \sum_l \mathbf{x}_l, \frac{1}{N} \sum_l \mathbf{x}_l^{\text{ref}}$
2: $\mathbf{x}_l, \mathbf{x}_l^{\text{ref}} \leftarrow \mathbf{x}_l - \mu, \mathbf{x}_l^{\text{ref}} - \mu^{\text{ref}}$
3: $U, V \leftarrow \texttt{torch.svd}(\sum_l \mathbf{x}_l^{\text{ref}} \otimes \mathbf{x}_l)$
4: $R \leftarrow UV$
5: **if** $\texttt{torch.linalg.det}(R) < 0$ **then**
6:      $F \leftarrow \begin{bmatrix} 1 & 0 & 0 \\ 0 & 1 & 0 \\ 0 & 0 & -1 \end{bmatrix}$
7:      $R \leftarrow UFV$
8: **end if**
9: $\mathbf{x}_l^{\text{align}} \leftarrow R\mathbf{x}_l + \mu$
10: **Return** $\mathbf{x}_l^{\text{align}}$

Alternatively, an *on-policy* strategy could be used, where conformations are generated dynamically during training, with energy calculated on the fly. Such an approach would necessitate maintaining a replay buffer to store and reuse past samples, enabling more adaptive training based on real-time model predictions. We leave this exploration for future work.

**Backbone model.** We adopt PROTENIX (Chen et al., 2025), a PyTorch implementation of AlphaFold3, as the backbone architecture. For all ATLAS training targets, we pre-computed the PairFormer embeddings (a collection of `s_input`, `s_trunk`, `z_trunk`) using $N_{cycle} = 10$. During training and inference, we activate only the input embedder and diffusion module while leaving other components, such as the MSA module, PairFormer, Distogram head and confidence heads frozen. Additionally, no template module or template-based input is used in our study. During training, the atom coordinates are properly permuted to match the best alignment with ground truth structure (Chen et al., 2025).

**Supervised fine-tuning (SFT).** Before performing EBA alignment, the pre-trained diffusion model is first fine-tuned using the standard score matching objective (Ho et al., 2020). The parameter optimization is performed with the Adam optimizer (Kingma & Ba, 2014), using a learning rate of 0.001, $\beta$ values of (0.9, 0.95), and a weight decay of $1 \times 10^{-8}$. The learning rate follows an ExponentialLR schedule with a warm-up phase of 200 steps and a decay factor $\gamma$ of 0.95 applied every 50k optimizer steps. We set $\lambda_{MSE} = 1.0$ and $\lambda_{LDDT} = 1.0$ in Eq. (14). During training the noise level is sampled from $\sigma_{data}e^{-1.2+1.5\cdot\mathcal{N}(0,1)}$ as the default setting with $\sigma_{data} = 16$. In each optimizer step, we clip the gradient norm by 10. The SFT process consists of two stages: in the first stage, input structures with more than 384 residues are randomly cropped to a fixed size of 384. For cropping, half of the time, contiguous (on sequence) cropping is used, while the other half employs the spatial cropping (Abramson et al., 2024). Random rigid augmentation is applied during diffusion training with an internal diffusion batch size of 32. In the second stage, the cropping size is increased to 768, and random rigid augmentation is applied with a reduced internal batch size of 16. In both stages, conformation samples are uniformly drawn from the training dataset without any sample weighting. The training was conducted with NVIDIA A100 GPUs.

**Alignment fine-tuning.** As an alignment baseline, the EBA-DPO is implemented within the EBA framework by setting $K = 2$ and replacing the softmax with sigmoid function as in Wallace et al. (2024). The binary preference label is annotated by selecting the sample with smaller energy in the mini-batch as "win" while the other as "lose". The reference model is selected to be the SFT model with all parameters frozen. For EBA, we follow the same optimizer and scheduler as SFT stage but use a smaller base learning rate of $1.0 \times 10^{-7}$. To reduce the variance of gradient during training, we accumulate the gradient per 16 steps and also clip the norm by 10. We set the energy temperature factor $\beta = 1/L^{0.5}$ where $L$ is the protein length of the current mini-batch of samples, and the combined model temperature factor $\alpha T = 50$. The (internal) diffusion batch size is set to be 8 during alignment. To obtain a batch of sequence-coupled samples, we first sample a single structure $(\mathbf{c}, \mathbf{x})$ uniformly from the alignment dataset, then we retrieve using the corresponding sequence $\mathbf{c}$ against the dataset for additional $K - 1$ sample with the same sequence $\mathbf{c}$ (uniformly sample from the dataset). Similar to SFT, we set $\lambda_{mse} = 1.0$ and $\lambda_{lddt} = 1.0$ as the structural loss weights. Similarly, experiments were run on NVIDIA A100 GPUs.

**Energy annotation.** To evaluate the potential energy $E$ of protein structures for preference fine-tuning, we adopted the OpenMM suite (Eastman et al., 2023) and CHARMM36 (Best et al., 2012) force field with GBn2 model (Nguyen et al., 2013) for protein solvation. Each protein structure was fixed by the PDBFixer to add hydrogen atoms and neutralized with $Na^+/Cl^-$ ions at a concentration of 150mM. We then perform energy minimization with tolerance as $10 \, kJ \, /(mol \cdot nm)$ until converge. The energy (unit in kJ / mol) after minimization was assigned to each structure. To calculate the Boltzmann weight, we adopted $k_B = 8.314 \times 10^{-3} \, kJ/(mol \cdot K)$ and set the temperature to be 300K (room temperature). Note that our energy evaluation does not perfectly reproduce the ATLAS simulation protocol, which uses GROMACS v2019.4 (Abraham et al., 2015) with the CHARMM36m force field (Huang et al., 2017) in an explicit TIP3P water environment. Since protein–solvent interactions can significantly contribute to the overall potential energy, using explicit water for energy annotation can introduce large variance in energy estimates and is far less efficient in computation. Future work may consider quantum-level single-point energy evaluation using density functional theory (DFT) methods, semi-empirical methods (such as GFN2-xTB (Bannwarth et al., 2019)) or even *ab initio* computational methods suitable for protein structure, which we believe is more promising since they are more accurate and less affected by solvent.

**Runtime profiling.** The MD ensemble generation runtime in Table 1 for baselines models (Jing et al., 2024a;b) and EBA are benchmarked on NVIDIA A100 GPUs. The training iteration per step and memory profiling in Table 2 are calculated based on $4\times$ A100 GPUs as wall clock time and per-device memeory consumption, respectively.

