# OpenReview forum: "Aligning Protein Conformation Ensemble Generation with Physical Feedback"
_ICML.cc/2025/Conference — ICML 2025 poster_

### Official Review · Reviewer_PSvo · 2025-02-23

**Overall Recommendation:** 3

**Summary:**

his paper introduces Energy-based Alignment (EBA), a novel approach that enhances generative models by incorporating feedback from physical models. EBA efficiently calibrates these models to balance conformational states based on energy differences, overcoming intractable optimization issues.

Tested on the MD ensemble benchmark, EBA achieves state-of-the-art performance in generating high-quality protein ensembles. By enhancing the physical plausibility of predicted structures, EBA improves model accuracy and shows potential for advancing structural biology and drug discovery applications.

**Claims And Evidence:**

Introduction of Energy-based Alignment (EBA):

Claim: The paper introduces the EBA method to enhance protein conformation ensemble generation by integrating physical energy feedback with generative models.
Evidence: The authors compare their method with traditional molecular dynamics (MD) simulations and generative models, demonstrating that EBA improves the physical plausibility of generated structures.
Validation of EBA's Performance:

Claim: The EBA-aligned model outperforms existing generative models in terms of generating high-quality protein ensembles.
Evidence: Experimental results on the ATLAS MD ensemble benchmark show superior performance in metrics like RMSD, RMSF, and exposed residue similarity compared to other methods (e.g., AlphaFold, AlphaFlow, and MDGen). The paper provides a table with detailed statistical metrics to support these claims​
.
Theoretical Justification of EBA:

Claim: EBA effectively aligns the model’s learned distribution with the Boltzmann distribution, ensuring better alignment with physical laws governing protein stability.
Evidence: The derivation of the EBA objective and its mathematical formulation (e.g., Equation 6 and 7) demonstrate how the method avoids the intractable partition function while still maintaining energy consistency across conformations​
.
Ablation Study on Hyperparameters:

Claim: The model's performance is robust across different values of the mini-batch size (K) during fine-tuning.
Evidence: The paper presents an ablation study showing consistent results for various K values, with only a slight increase in computational overhead for larger values​
.
Limitations:

Claim: The method has limitations, such as challenges with modeling long-time scale dynamics and the accuracy of the energy functions.
Evidence: The authors acknowledge that while the method works well for single-chain protein ensembles, further improvements are needed, especially in terms of force field precision and extending the method to more complex biomolecular systems​

**Essential References Not Discussed:**

Yes, some critical related works are not discussed.

Please see: Protein Conformation Generation via Force-Guided SE(3) Diffusion Models (ICML24) and the related works part.

In the submission, the authors did not have a paragraph for related works on protein conformation generation. Moreover, the baselines are not compared.

**Experimental Designs Or Analyses:**

Comparison with Baseline Methods:
Experimental Design: The authors compare their EBA-aligned model to existing protein structure generation methods, including AlphaFold2, AlphaFlow, and MDGen. These comparisons are made using the ATLAS MD ensemble benchmark, which involves generating protein ensembles and assessing their physical plausibility and other properties like RMSD, RMSF, and Jaccard similarity.
Validity Check: The use of established baselines and the clear presentation of comparative metrics (e.g., RMSD, RMSF, and other structural observables) helps establish the validity of their approach. The experimental results are robust, with the EBA-aligned model showing consistent improvements in key metrics like RMSF correlation and exposure of residues.
Potential Issue: The performance of baseline methods like AlphaFold2 and MDGen on different datasets or settings could impact the generalizability of the results. For example, if these baselines were trained or optimized differently, comparisons may not fully reflect the potential of the proposed method. This could be a source of bias.

Dataset and Training Pipeline:
Experimental Design: The training pipeline involves fine-tuning a pre-trained denoising network on protein ensembles from the ATLAS dataset. This ensures that the model learns from simulation data that captures diverse conformational states.
Validity Check: Using the ATLAS dataset, which consists of MD simulation data with atomistic resolution, is a good way to validate the approach against realistic, high-quality data. The dataset also covers a range of protein types, which improves the robustness of the results.
Potential Issue: The authors acknowledge that the training data is limited to single-chain proteins. Extending the analysis to multi-chain or larger protein systems could reveal the limitations of the model. Furthermore, while ATLAS is a useful dataset, the findings may not fully translate to other datasets without similar structural diversity or complexity.

**Methods And Evaluation Criteria:**

Yes, the methods and evaluation criteria makes sense to me.

**Other Comments Or Suggestions:**

Please add one paragraph in the related work to discuss generative AI for conformation generation.

**Other Strengths And Weaknesses:**

A series of baseline methods are not compared, including but not limited toEigenFold, Str2Str, ConfDiff, BioEmu, and AI2BMD.

**Questions For Authors:**

The evaluation was conducted on the ATLAS MD ensemble dataset. Have you tested the model on other protein datasets with different properties or larger systems, such as those involving membrane proteins or disordered proteins? If not, do you anticipate any challenges in transferring the model to these types of datasets?

In your ablation study, the results indicate that increasing the mini-batch size (K) does not drastically affect the performance but increases computational overhead. Can you quantify the computational trade-offs in terms of training time and hardware requirements for different K values? How scalable is the method for much larger datasets or systems with tens of thousands of atoms?

While the paper shows promising results against empirical MD simulation data, how do you plan to validate your approach against quantum-level calculations, especially for proteins with highly intricate folding mechanisms? Are you planning any future benchmarks that involve comparing your method to quantum chemistry methods, such as those based on density functional theory (DFT)?

Your current work is focused on generating ensembles for single-chain proteins. How do you plan to extend the model to handle multi-chain or large protein complexes? Are there any expected challenges or changes in the model architecture when scaling to multi-chain systems?

**Relation To Broader Scientific Literature:**

Yes, key contributions of the paper rare elated to the broader scientific literature

**Theoretical Claims:**

Yes, the proofs are mainly correct.

---

> ### Author Rebuttal · Authors · 2025-03-31
>
> We sincerely appreciate the thoughtful feedback and helpful suggestions from the reviewer PSvo. We here address each point raised and clarify aspects of our work where necessary.
> 1. Additional baselines. We acknowledge the importance of comprehensive baseline comparisons. In our study, we primarily focused on methods that directly optimize for MD ensembles with specific parameters (temperature, force field, etc.). Though the mentioned methods are all pioneering and relevant works, most of them are not trained on the Atlas data so we think it is not very suitable to compare with. Yet we are happy to include these baselines (with necessary MD fine-tuning) and set up a benchmark in the near future. We here complement the most recent two, Confdiff and BioEmu to the comparison:
>
> |Baseline| Pairwise RMSD r | Global RMSF r | Per target RMSF r | RMWD | RMWD trans | RMWD var | MD PCA W2 | Joint PCA W2 | PC sim > 0.5 \% | Weak contacts J | Transient contacts J | Exposed residue J | Exposed MI matrix rho |
> |----|----|----|----|----|----|----|----|----|----|----|----|----|----|
> | BioEmu | 0.46 |0.57 | 0.71  | 4.32 | 4.04 | 1.77 | 1.97 | 3.98 | 51 | 0.33 | - | - | 0.07 |
> |ConfDiff (OpenFold-r3-MD)| 0.59 | 0.67 |0.85|2.76|2.23|1.40|1.44|2.25| 35 |0.59|0.36| 0.5 | 0.24 |
> |EBA(Ours)| 0.62|0.71|0.90|2.43|2.03|1.20|1.19|2.04|44|0.65|0.41|0.70|0.36|
>
> We shall add these results to the paper after revision, thanks!
>
> 2. Related work section on *Generative AI for Protein Conformation Generation*. We are sorry if we don’t make it clear. The current manuscript does contain a paragraph for protein conformation generation in Section 5-Part 1, which includes several related works but may not sufficient. To provide a better context, we will make a more dedicated discussion on this topic in the revision, especially with the aforementioned baselines.
> 3. **Questions**. We kindly appreciate each of the reviewer's insightful questions, and we answer each as follows:
>
> **Q1**: By scaling up to larger systems such as membrane proteins or IDP, the network may suffer from capacity issues and lack of accuracy. 1. The training step (of AF2, AF3-like) is based on a cropped region of original structure (eg. 384, 768) which may be difficult in accurately predicting the distribution of larger protein. 2. AF3 is pretrained on structural proteins mostly and thus the sampled ensemble has limited diversity; the distribution of IDP is more flat in 3D space so the model should be finetuned to be able to cover such “free” ensembles.
>
> **Q2**: Sure. We conduct a simple benchmarking on different K (2,3,5) regarding training speed and GPU memory utilization. We use 4xA100 for profiling:
>
> | K | Iteration per step (sec) | Avg memory consumed in MiB (DDP, per device) |
> |--------|----------------------------------|---------------------------------|
> | 2  | 4.3                          | 12288.0                              |
> | 3  | 5.4                          | 14254.1                              |
> | 5  | 7.8                          | 16711.7                              |
>
> As we adopt the backbone of Protenix, the training of larger system (eg. >10k atoms or ~1k tokens) will be trained with a random cropping at a feasible training size (Eg. 384) on common A100 GPUs. In our opinion, it is doable in scaling up, while we acknowledge the challenge in accuracy for longer protein may exist due to the cropping, which is the shared issue for all structure prediction models.
>
> **Q3**: Yes. For the quantum-level, we plan to first experiment with the small molecule/peptide dataset. For example, we plan to curate a molecular conformation dataset annotated by the gfn2-xtb, a well-known semi-empirical method. For the protein, the quantum calculation is not computationally feasible; we may consider the estimated potential such as the one in the AI2BMD paper. While compared with quantum-level methods would provide additional insights, such approaches are computationally expensive and are generally limited to small peptides rather than protein targets from ATLAS at similar time scales. We leave for future exploration.
>
> **Q4**: Yes, in fact extending to multi-chain (complex) is right on our way. We are actively experimenting to extend the EBA to the protein-ligand simulation data (specifically, the MISATO dataset (Siebenmorgen et al., 2024)). Since the AF3/Protenix architecture by design can handle complex input, we think it is ready w.r.t. model architecture. We also anticipate that this will require modifications to the energy function to account for the inter-chain interactions.
>
> We sincerely appreciate the insightful comments and questions, which have helped us identify areas for clarification and improvement. We hope that these revisions will address your concerns and further strengthen our manuscript.

---

### Official Review · Reviewer_yVMH · 2025-03-08

**Overall Recommendation:** 4

**Summary:**

This work presents Energy-based Alignment (EBA), a method to fine-tune a pretrained diffusion model to sample mini-batches of protein structures that match the underlying Boltzmann distribution.  This alignment is achieved by minimizing the cross entropy between the ground truth Boltzmann distribution, and the distribution approximated by the model, evaluated on the mini batches of size K. Authors provide a theoretical derivation of the exact form of the cross entropy for the diffusion model. Additionally, authors show that EBA with K=2, in the low temperature limit, can be formulated as a Direct Preference Optimisation problem (EBA-DPO).

Authors fine-tune AlphaFold 3 (Protenix) on the ATLAS dataset. To compare against baselines, authors use metrics from Jing et al. 2024. Both EBA and EBA-DPO outperform baselines across almost all metrics.

**Claims And Evidence:**

Authors claim in the abstract that “Experimental results on the MD ensemble benchmark demonstrate that EBA achieves state-of-the-art performance in generating high-quality protein ensembles”, and that indeed seems to be supported by the experimental evaluation. AlphaFlow-MD, MSA subsampling and MDGen are strong baselines that EBA outperforms.

**Essential References Not Discussed:**

While the presented methodology is original, it is based on previous findings that are only shortly discussed. Boltzmann generators from Noe et al. 2019 (https://arxiv.org/abs/1812.01729) are based on the similar insights about matching equilibrium distributions, and if they are the inspiration for this work, they should receive more attention.

**Experimental Designs Or Analyses:**

Using AlphaFold 3 (via Protenix) is a good choice of a model for the task at hand. It is not clear what was the training (fine-tuning) procedure (for example, optimizers, or expanded discussion of training instabilities from line 338).

**Methods And Evaluation Criteria:**

ATLAS dataset is a commonly used dataset for this task, and the chosen baselines make sense. The generation success is assessed using metrics commonly used in similar works.

**Other Comments Or Suggestions:**

In Section 2, where DPO is introduced for the first time, ${x^w}$ and ${x^l}$ are not explained. It would improve reading of the work if more intuition about BT model was given.

In Algorithm 1, step 10, the comment is maybe a bit close to the text and looks like a division.

Small typo: line 109, ‘that is ,’ instead of ‘that is, ‘

Main body, rather than the Appendix, could mention how the energy function ${E}$ is chosen.

**Other Strengths And Weaknesses:**

Strengths:

The paper is generally well written and has a solid evaluation. Including the pre-trained version of the model, but before EBA fine-tuning, shows evidence that EBA indeed results in improved performance.

Weaknesses:

Authors don’t discuss properly in the Methodology Section what is their own novel idea, and what is inspired by other works.

**Questions For Authors:**

1.	Is the smooth LDDT loss an entirely novel thing introduced here, or it is taken from other works?

2.	In Algorithm 1, in step 3., why $K-1$ samples are retrieved? Does $c^j = c(j=…)$ simply mean that enumeration?

3.	From reading of the manuscript it seems that eta is the learning rate, and lambda is the loss weight. Are those the same eta and lambda as in the Figure 4?

**Relation To Broader Scientific Literature:**

The problem that this work is addressing, namely sampling from the ensemble that obeys the ground truth Boltzmann distribution, is a known and important problem in structural biology, see for example a review by Henin et al. 2022 (https://arxiv.org/pdf/2202.04164)

**Theoretical Claims:**

I went through derivations in the main body, but didn’t check the Appendix. Authors optimise ELBO using previously derived results, however the substitution of the denoising loss into KL divergence (line 185) should be made more explicit.

---

> ### Author Rebuttal · Authors · 2025-03-31
>
> We kindly appreciate the reviewer yVMH’s supportive and insightful feedback. Below, we address each of the key questions raised and provide clarifications where necessary.
> 1. Clarifications on theoretical derivation. We acknowledge that the substitution of the denoising loss into the KL divergence (Line 185) was not made fully explicit. In the revised manuscript, we will provide a more detailed explanation of this substitution and its connection to ELBO. Specifically, we will highlight the derivation steps that lead to the energy-based alignment (EBA) objective and its direct connection to narrow the gap towards the Boltzmann distribution.
> 2. Training details and discussion. Thanks for pointing out the suggestion for clarification.
>
> [Training details] In the current paper version, we actually postponed the implementation details in the appendix to save space in the main text. To clarify: we used Adam as the optimizer with a learning rate of $10^{-3}$ during SFT and $10^{-7}$ during EBA, $\beta$ values of (0.9, 0.95), and a weight decay of $1×10^{−8}$. The learning rate follows an ExponentialLR schedule with a warm-up phase of 200 steps and a decay factor $\gamma$ of 0.95 applied every 50k optimizer steps. Gradient norm clipping was applied to each step by 10;
>
> [Line 338] The training instabilities were primarily observed early that when we did not impose the length factor $1/L^{0.5}$, the softmax of energy would spike. This is due to the change of scale of energy. Traditional molecular mechanism force field (MMFF) is defined by enumerating the interactions existing in the molecule. The increase of atom number (or protein length L) will (quadratically) also increase the pairwise terms. One can imagine that in large proteins, the energy scale E(x) is much larger, such that the energy difference [E(xi) - E(xj)] is thus linearly larger, which makes the learning difficult for large systems since the energy softmax can spike instead of being balanced between conformational states.
>
> We will expand on these points and organically incorporate them into the main text in the revised version to provide more transparency on the training process.
>
> 3. Discussion on prior works and clarification on our novel contributions. Thanks for the good suggestions. We acknowledge that Boltzmann generators (BG, Noé et al., 2019) is a very pioneering work in this research direction. While our approach differs here in using diffusion models and model alignment for fine-tuning, we agree that the core principle of learning towards the Boltzmann weighted ensembles is shared. We were so inspired by many prior works including BG, RLHF, DPO, AlphaFlow to name a few, and will dedicate a discussion paragraph in the introduction section to motivate the EBA more explicitly.
> 4. Tentative summary of novel contributions. We acknowledge that the methodology section does not clearly separate our novel contributions from the prior works. As for our contributions, we showcase an successful application of fine-tuning AF3 for MD ensemble generation. Also:
> - we shall explicitly state that EBA introduces a new physical-inspired fine-tuning objective based on energies for diffusion models, aligning generated distributions with the Boltzmann distribution.
> - we will clarify that the connection to DPO is novel, as previous works have not formulated preference optimization within a Boltzmann distribution motivated framework.
> - the use of mini-batch Boltzmann weighting for preference optimization is also a novel aspect of our work to enable scalable training of models such as AF3.
>
> In conclusion, we will re-structure the methodology section to better highlight these contributions and make clear which points are inspired by the pioneering works.
> 5. Responses to the specific questions:
>
> [**Q1**] Smooth LDDT Loss: The smooth LDDT loss is inspired from structural alignment objectives in training the diffusion module of AlphaFold3, and is so used for diffusion fine-tuning in our work. We will clarify this in the manuscript.
>
> [**Q2**] Retrieving K-1 Samples: K-1 indicates the extra samples additional to the original training sample i. In this sense, K=2 means we would retrieve 1 extra related sample to compute the energy weight; Yes, this notation simply denotes enumeration over samples. We will revise this for clarity.
>
> [**Q3**]Learning rate eta and loss weight lambda: We are sorry for abusing the usage of these letters, they do have different meanings: in the Figure 4, these symbols are introduced from the original AF3; while in the algorithm 1 they are learning rate and loss weight. Subsequently, we will rename them and clarify this explicitly in the text.
>
> We thank the reviewer yVMH for their positive evaluation of our work and for recognizing the contributions of EBA. We will incorporate these suggestions to improve clarity, expand discussion on related work, and provide additional experimental details.

---

### Official Review · Reviewer_BE4T · 2025-03-14

**Overall Recommendation:** 3

**Summary:**

This work concerns the problem of improving diffusion models for protein conformation generation using the information from a physical model. It proposes a new fine-tuning loss EBA for diffusion model training, based on the principle of offline RL using energy labels as the a negative reward feedback.

By balancing the diffusion loss for different conformations of a protein during mini-batch training, it aims to train the model to better reflect the thermodynamics of the protein. In addition, the authors show that the popular DPO objective is a special case of EBA, by using a mini-batch of 2 and setting an infinitely low temperature. Through the evaluation on the standard benchmark of Atlas, they shows that models fine-tuned using EBA objective achieve better performance in capturing the flexibility of the protein and predicting the dynamics-related MD observables.

**Claims And Evidence:**

The main claims made by the authors includes:

1. “Our approach bridges the gap between purely data-driven conditional generation and physics-based simulations by incorporating fine-grained force field feedback via a scalable learning objective”
2. “Through the EBA alignment, the diffusion model learns to balance different conformational states, resulting in more physically consistent protein conformation ensembles”
3. “the EBA-aligned diffusion model achieves state-of-the-art performance compare to previous generative models by incorporating physical feedback”
4. “the proposed method provides a novel pathway for integrating generative modeling with transitional simulations”

While claim 1 and 4 are supported by the construction of the problem, questions remain for claim 2 on how does the learned diffusion model align with the target Boltzmann distribution (Q1); for claim 3 on the practical improvement of using EBA over SFT-fine-tuned model (Q3).

**Essential References Not Discussed:**

Recent efforts have attempted to incorporate physical information during the training and sampling of diffusion models to improve the protein conformation generation. For example:

ConfDiff (ICML 2024, https://arxiv.org/abs/2403.14088) trains plug-in guidance models to integrate energy and force information during the reverse sampling. It also targets the energy-tilted distribution (eq 3 in their paper) and connects adjusted reverse diffusion with the potential energy and force of conformations (eq 4, eq 5 in their paper).

BioEmu (preprint, https://www.biorxiv.org/content/10.1101/2024.12.05.626885v1) proposes to fine-tune pre-trained diffusion models with the property prediction fine-tuning, integrating the energy difference of pairs of conformation states (Section S3.6 in their paper).

These works should be discussed in order to better understand the current approach.

**Experimental Designs Or Analyses:**

As proposed method is a new objectives for diffusion fine-tuning, one key comparison should be proposed method vs SFT, which is missing from the main result (see Q3).

**Methods And Evaluation Criteria:**

See Q1, Q2, Q5 regarding the questions on the proposed methods.

The evaluation criteria is standard in the field and is appropriate for the proposed problem.

**Other Comments Or Suggestions:**

There are several notation issues and typos in the current manuscript:

- Equation (1): what is $\lambda_t$  in $\omega(\lambda_t)$?

- In Equation (2), using $\theta$ for both $r_\theta$ and $p_\theta$ can be misleading that i) two models share parameters and ii) reward model is also optimized in equation (2)

- Equation (6):
  - There is no superscript "i" in first summation and there is a missing “i” in $\log(p_\theta(x|c))$
  - The summation is a Monte Carlo estimation of the KL divergence, yet a equal-by-define $\triangleq$ instead of approximation $\approx$
  - In Line 162: used summation over $x^i$

- Typo: Line 246 right column: “predicted coordinates $\mathbf{x}_0$”
- Typo: Line 608: mixed use of index i and j
- Equation 17, additional comma ‘,’ in the denominator
- In Section A.1.: are $\forall j$ and $\forall i$ necessary in some of equations?

**Other Strengths And Weaknesses:**

- Strength: this work shows a new training objective to integrate protein conformation energy for diffusion model training, using AlphaFold3 as a modern all-atom framework, and empirical results show strong performance on Atlas benchmark.
- Weakness: my main concerns originate from some specific details of the model (see questions). In addition, there is an excessive mixed use of notations (see Other Comments)

**Questions For Authors:**

1. Can you clarify which target distribution the model learns to match for the general EBA objective (Equation 7) and for the model trained in Table 1? Based on the model, it should match to a weighted distribution of data $\propto p_\text{data}(x)\exp(-\beta E(x))$, however, the data from MD simulation already follows the Boltzmann distribution of the potential energy $\mathbf{x}_\text{data} \propto \exp(-\beta E(x))$. Imposing this additional energy weights seems to bias the target distribution away from the Boltzmann distribution. Can you provide further clarification?
2. In equation 11 and 12, why is the total number of diffusion steps (T) in the exponential part of the denominator instead of before the summation?
3. What is the base performance for SFT model, after the training stage 1?
4. At first, It appears that a large K value is required for an accurate Monte Carlo estimation of the partition function in Equation 7; however, the ablation study in Table 2 suggests that the results are not sensitive to the choice of K. Can you explain the principle behind selecting an appropriate K value?
5. I have several questions regarding the treatment of potential energy E(x):
    1. The authors mentioned that E(x) varies significantly with protein size, introducing large variance in the objective. Can you clarify why this is an issue, given that all data are normalized by mini-batch Boltzmann weights (Algorithm 1, step 5) and are from the same protein (i.e., same protein size)?
    2. By using length dependent normalization, the proposed method appears to impose different temperatures for different proteins. Can you clarify if this is the case, and if so, what potential issues may arise from this treatment? Additionally, can you explain why “the scaling of folding time with respect to the length” translates into the rule of scaling the potential energy with the length?
    3. Large energy variance can still exist within different conformations of the same protein and cause the softmax operation in Algorithm 1, step 5 to collapse  — only one or a few data have probability ~1 and others to be 0. Did you check the mini-batch Boltzmann weights across different proteins?
6. In Appendix A.2, the authors “combine alpha and T” into a single hyperparameter, presumedly the “model factor T” mentioned in Appendix B (Alignment fine-tuning). This combination is somewhat confusing. For exmple, when varying the “model factor T”, did the authors also change the number of diffusion steps (the original T)?
7. While using structural losses specific to protein (aligned MSE and smooth LDDT) can be a reasonable alternative for training in practice, the reasoning is not entirely convincing: the use of MSE error of predicted noise is derived from the KL divergence of stepwise reverse Gaussian steps whereas the structural losses may not. Is there any approximation introduced by replacing with proposed structural losses? Did you conduct empirical studies on choosing different losses?

**Relation To Broader Scientific Literature:**

The general problem of this work is to improve a pre-trained diffusion model with an explicit reward feedback (i.e., energy), with a specific focus on distributional matching. This general problem was discussed in a recent paper Lu et al (ICML 2023, https://arxiv.org/abs/2304.12824) with an extension to protein conformation (Wang et al, ICML 2024, https://arxiv.org/abs/2403.14088). However, above work approaches the problem from the perspective of guidance while this work attacks problem by forming a new fine-tuning objective.

**Theoretical Claims:**

I read through the derivation and proofs of the paper. Most of the derivations are standard and resembles those in DPO and Diffusion-DPO (Wallace et al). See Q2 for a specific question.

---

> ### Author Rebuttal · Authors · 2025-03-31
>
> We sincerely appreciate the detailed and constructive feedback from the reviewer BE4T! We will address each question/concern as follows:
>
> 1. Target distribution and bias away. Indeed, given sufficient unbiased MD data, directly maximizing the likelihood (MLE) is ready to converge to the Boltzmann distribution (Noé 2019, Arts 2023). However, we note that the ATLAS MD dataset, containing "collections" of 3 parallel trajectories per target, may introduce bias when using MLE alone. In this context, EBA adjusts for potential bias by incorporating partial information from the Boltzmann factor.
> Furthermore, our objective function is unbiased towards $p(x)$ from the following sense: one can prove if there exists a distribution $q_\theta(x) = \exp(−β’ E_\theta (x)) / Z’$ such that $q(x_i)/q(x_j) = p(x_i)/p(x_j) \forall i, j$, then $p(x)$ equals to $q(x)$ for all x. This implies that optimizing w.r.t. the Boltzmann factor converges towards the target Boltzmann distribution. We will add this proof to the next revision to inspire better insights.
>
> 2. Eq. (11, 12) notation of T. It is a result of taking expectation of $t \sim U(0, T)$. During derivation, we make the original sum over diffusion time step $\sum_{t=0}^T$ in ELBO as $T * [(1/T) * \sum_{t=0}^T]$ and extract the second part [**] out using the Jensen Inequality to be there.
>
> 3. Base performance before EBA. This is indeed an important ablation study. Below is the evaluation results of the SFT checkpoint before EBA. We will add this into the next revision:
> |Name| Pairwise RMSD r | Global RMSF r | Per target RMSF r | RMWD | RMWD trans | RMWD var | MD PCA W2 | Joint PCA W2 | PC sim > 0.5 % | Weak contacts J | Transient contacts J | Exposed residue J | Exposed MI rho |
> |---|---|---|---|---|---|---|---|---|---|---|---|---|---|
> |Stage 2-SFT| 0.57|0.69|0.89|2.58|2.15|1.28|1.29|2.13|42|0.63|0.43|0.67|0.35|
>
> 4. Selecting K value. We note that using K samples in the EBA objective is not intended to estimate the partition function and likely cannot do so accurately. Instead, the EBA objective aligns with Boltzmann factors, i.e., the relative weights between conformations. Since these relative weights are invariant to the number of samples used, increasing K does not necessarily improve performance.
>
> 5. Treatments of potential energy E(x). We appreciate the insightful questions which we are happy to discuss as follows:
>
> (1) This is due to the change of scale of energy. Traditional molecular mechanism force field (MMFF) is defined by enumerating the interactions in the molecule. The increase of #atom (or protein length) will quadratically increase the pairwise terms. In large proteins, the scale of E(x) is much larger, such that the energy difference [E(xi) - E(xj)] is linearly larger, which will make the learning difficult for large systems since the energy softmax can spike instead of being balanced.
>
> (2) [i] Correct, following the point above, the introduction of length-factor implicitly changes the temperature of different proteins, thus calibrating the coefficient. The potential issue can be a push-away from the target relative weight between conformations yet the order of them is kept therein. Thus, EBA is inducing a much “flatter” distribution for large proteins.
>
> [ii] the folding time can be hypothesized as a result of free energy barriers. Inspired by their empirical findings (Naganathan, 2005), we thus applied the L^0.5 and found it empirically worked out fine. We cite this to make an analogy instead of being a rigorous scaling rule.
>
> (3) In fact, it is the problem of “energy variance” that motivates us to use the off-policy ensembles where the conformations embody smaller variance (due to being sampled from MD). As a result, the range of energy difference is small and the softmax weights stay within (1e-2, 1). In early experiments, we found that the softmax weights will spike (as small as 1e-7 vs 1.0) if we do not introduce the length factor, and even worse when we used on-policy data (sampled using pre-trained AF3).
>
> 6. Merged hyperparameters. We are sorry if this introduces ambiguity. In fact, this trick is inspired by the diffusion-DPO where the authors did the same combination. In practice, we only tune $\alpha * T$ as an independent variable, keeping the sampling diffusion step as T=20 for all configs.
>
> 7. Structural aligned loss. The rationale of using these losses come from the original training of AF3 ’s diffusion module, and we mostly follow their loss definition. The consequence is that the space of protein structure is not a quotient space R^{3N} / SE(3) according to AlphaFlow paper section 3.2 (Jing, 2024). We have not yet abated on the loss choices and will clarify this in the paper.
>
> 8. Additional related works, notation/typo, and formatting issues (comments): We are grateful for pointing out these valuable suggestions. We appreciate the attention to detail and will dedicate the discussion and the corrected issues into our next revision.

---

> > ### Comment · Reviewer_BE4T · 2025-04-05
> >
> > I thank the authors for clarifications on the model details and additional results. Here are some follow up questions and comments:
> >
> > 1. Can the authors clarify and provide the proof on which distribution the EBA objective (eq 7) learns given a proposal distribution $p*$? I am still not convinced that solution is unbiased toward the target Boltzmann distribution. Here is my thought process and please kindly let me know if I missed anything (for compactness, condition $\mathbf{c}$ is omitted and random variable $x$ is not **bolded**):
> >
> >     Based on eq (6), minimizing eq (7) using data $\\{x\\}$ from a proposal distribution $p*$ is equivalent to optimizing
> >     $$\mathcal{L}(\theta) = \mathbb{E}\_{x \sim p^*}[p\_B(x) \log \frac{p_B (x)}{p\_{\theta}(x)}] = \underbrace{\mathbb{E}_{x\sim p^*(x)}[p_B (x)\log{p_B (x)}]}\_{\text{constant}}  - \mathbb{E}\_{x\sim p^*(x)}[p\_B(x)\log{p\_{\theta}(x)}].$$
> >     Define a new distribution $r(x) = \frac{1}{Z\_r} p^*(x)p\_B(x), Z\_r=\int p^*(x)p\_B(x) dx$, the objective becomes
> >     $\mathcal{L}(\theta) = -Z\_r \mathbb{E}\_{x\sim r(x)} [\log p\_\theta(x)] + \text{constant}$, which is the MLE with respect to the **tilted distribution $r(x)$**.
> >
> > 2. Thanks for the clarification on the derivation of $T$. The equation in Appendix A.1 Line 641 should reflect the inclusion of $T$ in the exponential part before $F(x^j, t)$ and $1/T$ before the summation $\sum_{t=1}^T$. The remaining derivations should be updated with the correct inclusion of $T$.
> >
> > 3. Comparing eq (9) and the left hand side of eq (17), the authors substituted $\log p\_\theta(\mathbf{x}^i|\mathbf{c})$ with $-\sum_t \mathbb{E}\_{x\_i\sim q(\mathbf{x}\_t^i|\mathbf{x}\_0^i)}\mathbb{D}\_{\text{KL}}[q(\mathbf{x}^i\_{t-1}|\mathbf{x}\_{0, t}^{i})\|p\_\theta(\mathbf{x}\_{t-1}^j|\mathbf{x}\_t^i, \mathbf{c})]$. This is different from the "reverse decomposition" of the markov chain in Wallace et al  that $\log p(x_{0:T}) =\sum_{i=1}^{T} \log p(x_{t-1}|x_t)$. Can the authors provide more details on why the substitution is valid?
> >
> > 4. The new result on SFT and before EBA is arguably the most important baseline to compare in order to *correctly interpret the effect of EBA*. Based on this new results, the major improvement over the metrics seems from fine-tuning AlphaFold3-like models on MD data and the improvement from EBA alone, albeit marginal, is observed.
> >
> > I agree that integrating physical feedbacks into model training is an important problem, and the authors provide an interesting solution. However, my concerns regarding the clarity and rigor of the method (Q1/Q3) remains. The authors should clarify why the objective is unbiased or, at least, discuss any approximations involved and their potential effects.

---

> > > ### Author Response · Authors · 2025-04-06
> > >
> > > Dear reviewer BE4T:
> > >
> > > We appreciate your further feedback and positive recognition of our method. We are happy to discuss more based on the listed question/comments:
> > > 1. Thanks for the tentative derivations and interest. We want to clarify in short that there are indeed some approximations and target distribution can be a tilted one due to such approximation; otherwise, it is unbiased.
> > >
> > >     - In fact, Eq. (7) stands as the "cross-entropy"-like form but involve 3 distributions: $p_B, p_\theta, p^*$ (Boltzmann distribution, model distribution, data sampler). Starting from the K-categorical distribution $p^K$ for the energy-based model (we denote $f(x, E, b) = e^{-b E(x)}$; use a,b instead of $\alpha, \beta$ to save space; set K=2 for simplicity): $KL(p^K_B||p^K_\theta) = - E_{x_1, x_2 \sim p_B} \sum_{i=1,2} \frac{f(x_i, E, b)}{f(x_1, E, b) + f(x_2, E, b)} \log \frac{f(x_i, E_\theta, a)}{f(x_1, E_\theta, a) + f(x_2, E_\theta, a)}$ (cross entropy term). Direct unbiased sampling from $p_B$ is intractable (that's why we are here), and we thus **approximate** it by some proposal distribution $p^*$ rather than p_B (in our study, this is off-policy trajectory data) while keeping the objective.
> > >
> > >     - **Then to clarify**: our previous response is intended to imply, when $p^* \approx p_B$ if we tolerate such approximation, we show that minimization of $KL(p^K_B||p^K_\theta)$ can converge to a minimizer of $KL(p_B||p_\theta)$ in the following sense:
> > >
> > >         - (*Proof*) If $\forall x_i, x_j, ~p^K_B = p^K_\theta$, which implies that $f(x_i, E, b) / f(x_j, E, b) = f(x_i, E_\theta, a)/f(x_j, E_\theta, a)$. Then we see $- b[E(x_i) - E(x_j)] = - a[E_\theta(x_i) - E_\theta(x_j)]$, or equivalently $E_\theta(x_i)= \frac{b}{a} E(x_i) + [E_\theta(x_j) - \frac{b}{a} E(x_j)], \forall i,j$. Note that by marginalizing j, we see that $E_\theta(x_i) = \frac{b}{a} E(x_i) + Const, \forall i$. Finally we plug in, $p_\theta(x_i) = exp( - a E_\theta(x_i)) / Z = exp( - a (\frac{b}{a} E(x_i) + C)) / Z = exp( - b E(x_i) ) / Z’ = p_B (x_i)$. qed.
> > >
> > >     - Otherwise rigorously, the target distribution goes to the tilted $p_{target}=\frac{\exp(-\beta E(x)) \times p^*}{Z^*}$ w.r.t. cross-entropy (CE) objective, as you derived above.
> > >
> > > We will compile this helpful discussion into revision and re-clarify what our target distribution is.
> > >
> > > 2. We have re-checked and spotted the confusing part, which will be amended according to the reviewer's advice. Thanks for your suggestion.
> > >
> > > 3. We are sorry due to some skip of explanations. We first elaborate how we arrive there: inspired by [Wallace et al, Eq. (8) <-> Eq. (11)], we first re-write the objective in Eq. (9) based on the ELBO form in Eq. (10), which involves an implicit change of $E_\theta(x; c)$ in Eq. 8 similar to [Wallace et al, Eq. (9)] to include all the latent variables $x_{1:T}$. Such that we define $E_\theta(x; c) \triangleq -\frac{1}{\alpha} (E_{p_\theta(x_{1:T}|x_0)} [\log p_\theta (x_{0:T}|c)] +  \sum_t H(q_t) + \log Z)$ in Eq. (8), where the $H(q_t)$ is the entropy of $q(x_{t-1}|x_{0,t})$. The presence of $H(q_t)$ is due to the omit of reference distribution $p_{ref}$ compared to Diffusion-DPO. Hence, the $\log p_\theta (x^j)$ term in Eq. (9) is supposed to be $E_{p_\theta(x^j_{1:T}|x^j_0)} \log p_\theta (x^j_{0:T}) + H(q_t)$ for sample $x^j$. Secondly, we apply reverse decomposition and approximate the proposal using the true posterior $q(x_{t-1}| x_{0,t})$, which yields $\sum_{t} [E_{q(x_{t-1}|x_{0,t})} \log p_\theta (x_{t-1}|x_t) + H(q_t)] \triangleq \sum_{t} J(x)$. Finally, $F(x^j, t) \triangleq -KL(q||p_\theta) \equiv H(q_t) - CE(q_t, p_{\theta,t}) = J(x^j)$.
> > >
> > >     - **Remarks**: Our prior Eq. (8-9) were intended to start with a general case (beyond diffusion), while during the following derivation in Eq. (9) we forgot to align the corresponding definition. We will revise this carelessness afterward and hope this makes more sense now.
> > >
> > > 4. Thanks for the objective comments. We want to further note that (not to argue):
> > >     - From the data utilization, the **AF3[Stage2-SFT]** was supposed to be comparable to **AlphaFlow**, and it does seem within such expectation, and is much better for several metrics.
> > >      - In regard of Atlas dataset, the model can already learn quite much from doing MLE/SFT on the trajectories. The EBA fine-tuning injects "something further" beyond that, so we see an improvement.
> > >
> > > We sincerely thank you again for the insightful suggestions, from which we have been aware of the unclarity therein. We would be grateful if you would consider raising your score in case we have addressed your concerns.

---

### Official Review · Reviewer_Gaxj · 2025-03-16

**Overall Recommendation:** 2

**Summary:**

This paper tackles the task of conformational ensemble generation in protein structure prediction. While folding models like AlphaFold predict individual states given a sequence, the task here is to be able to sample from the entire Boltzmann distribution instead. To this end, the authors propose an alignment scheme that leverages physical energies from molecular force fields to align the diffusion model component of modern folding models like AlphaFold3 to sample from the Boltzmann distribution. In particular, the authors introduce a scheme that effectively weighs different samples from the distribution according to their Boltzmann factors when fine-tuning the model, related to reward-based fine-tuning methods. The model is trained on molecular dynamics data that offers approximate samples from the Boltzmann distribution for training and compared to recent works such as AlphaFlow and MDGen, achieving favourable performance.

**Claims And Evidence:**

The main claims made by the submission are generally supported through empirical evidence presented in the experiments.

**Essential References Not Discussed:**

I think all most essential papers are cited.

However, when deriving its alignment scheme, the paper employs an energy-based formulation, essentially starting its derivation from a forward KL minimization (equation (6)). The method then approximates the partition function through a small set of empirical samples, if I am understanding the paper correctly (in practice, it is merely 2 samples in the main experiments). This raises questions, which I will discuss below. With that in mind, it would be appropriate here to cite works from the broader literature on energy-based models and position the method appropriately, contrasting it to other approaches. See, for instance, Song and Kingma, *How to Train Your Energy-Based Models* https://arxiv.org/abs/2101.03288, 2021, for an overview and further citations therein to more methods (contrastive divergence, noise contrastive estimation, Stein discrepancy, etc.). Among other things, these methods usually require a careful handling of the partition function, which in practice can require the need for MCMC and similar methods, but this is not the case in the submission here.

**Experimental Designs Or Analyses:**

Experimental design and analyses overall seem appropriate and seem sound an valid. I do have some questions regarding the experimental results in an ablation experiments, though. See below.

**Methods And Evaluation Criteria:**

The methods and evaluation criteria are generally appropriate and make sense for the problem at hand. I do have some concerns regarding the presentation and derivation of the newly proposed method, though. See below.

**Other Comments Or Suggestions:**

- Line 090: The equation seems incorrect. For $q$, the authors have the transition from $x_{t-1}$ to $x_t$, but then they write the Normal kernel that goes from clean data all the way to $x_t$.
- For consistency, the authors should make the connection between $x_\theta$ and $\epsilon_\theta$ in the background section.
- Below equation (14), I assume it should be *predicted coordinates* $\hat{x}_0$ with the hat?
- Figure 3 is missing the x axis description and annotation.
- Figure 4, last 2 plots, are missing the y axis description.

**Other Strengths And Weaknesses:**

**Strengths:**
- The empirical results in the main table, although somewhat incremental, show the proposed method's advantage.
- Generally, the approach to incorporate explicit physical energies based on atomistic force fields when doing generative modeling of conformational ensembles makes sense to me, so I think this is a worthwhile direction.

**Weaknesses:**
- In their derivations, the authors simply approximate the partition function through a small set of samples from the training data (a mini batch, or just two samples, i.e. K=2). I think this is a very, very coarse approximation of the true partition function. However, I think this not only introduces variance in the partition function estimate, but also a strong bias relative to what one would obtain with the ground-truth partition function. Usually, when training energy-based models, it is this partition function that leads to many complications. This approximation and the potential bias is not discussed at all and completely ignored. I would like the authors to comment on that.
- One would think that using more samples, i.e. a higher K, to approximate the partition function would be more accurate and consequently lead to better results. But this does not seem to be the case. The ablation study seems to indicate that the method is not very sensitive to K and that for some metrics the results actually get worse. Again, this raises questions, but is not discussed at all.
- The way the method is derived is confusing: In the background section, in equation (2), the authors first introduce $\alpha$ as the regularization strength in RLHF. Then above equation (6), $\alpha$ is used again, but this time as a scaling of the energy, which is different from the previous $\alpha$ and in fact would not be necessary at all, as it could just be absorbed into the energy function definition. Moreover, after equation (8), the $\alpha$ effectively cancels out and there is no $\alpha$ anymore in equation (12). Then, in section 3.3 the $\alpha$ re-appears, and now it is reinterpreted as the regularization-$\alpha$ from RLHF. This approach is not incorrect, but it is confusing. It would make a lot more sense to not have the second $\alpha$ introduced above equation (6) at all, and instead in section 3.3 when making the connection to DPO to define $E_\theta=-\log (\frac{p_\theta}{p_{ref}})^\alpha$. This makes intuitively much more sense, because now it is clear that this is again the regularization $\alpha$ that modulates the fine-tuned $p_\theta$ with respect to $p_{ref}$. Overall, I thought the derivation, although mathematically correct, was quite confusing.
- In the main experiment table, it seems that EBA-DPO performs almost on-par with EBA. Hence, the advantage of the full EBA framework over regular DPO is very incremental.

The above concerns do impact the paper's clarity and significance, unfortunately.

**Questions For Authors:**

I do not have any further questions.

**Relation To Broader Scientific Literature:**

The key contributions of the paper are positioned appropriately with respect to the existing literature. In particular, the paper appropriately references and discusses the most important protein folding models, methods to perform ensemble generation instead, as well as the most important alignment and fine-tuning methods.

**Theoretical Claims:**

Yes, I checked the correctness of the proofs in the appendix. Everything looks correct to me.

---

> ### Author Rebuttal · Authors · 2025-03-31
>
> We sincerely appreciate the reviewer Gaxj's detailed feedback and constructive criticism. Below, we address the main concerns raised in the review and clarify points of potential confusions.
> 1. Clarification of the learning objective. The reviewer raises a point regarding the “approximation of the partition function Z” using a small set of samples (eg., K=2) and the potential biases introduced therein. There have been works exploring effective estimation of intractable partition function and the induced unbiased sampling; however, we do not claim such. We acknowledge that approximating Z with limited samples introduces large variance, but we emphasize that our method is motivated by/focuses on aligning with the relative Boltzmann factor (BF, p(xi)/p(xj)) between the off-policy conformations as the learning objective. Indeed, EBA can be viewed as an alignment “towards” the Boltzmann distribution while using partial information from mini-batched samples. Our finetuning objective is "emulating" the Boltzmann-weighted distributions rather than precisely sampling from the Boltzmann distribution, to (ideally) make the model situate between the pre-training and Boltzmann weight-aware. To further clarify this, we will include extra discussion in the revised manuscript regarding the impact of K, along with a more in-depth explanation of how our approximation aligns with other energy-based modeling approaches.
> 2. Discussion on the sensitivity to K. The reviewer points out that the increased K which better approximates Z does not “consequently improve performance”. Our hypothesis is that the effectiveness of our method depends more on the distribution/quality of the off-policy structures rather than the size of the mini-batch. Firstly, K=2 does not imply there are totally 2 conformations sampled from the same target; different batches will roll over the acquired K samples and thus theoretically have access to each pair of p(xi)/p(xj). Secondly, we were adopting a simplified off-policy dataset from 100ns MD trajectories, where the dataloaders do not always yield a distinct variety of modes, since the structural fluctuations can be small for 100ns simulations. This may explain why performance does not always improve. To make readers aware of this concern, we shall expand our discussion on the interplay between K and performance, and such challenges probably anticipated in the on-policy setting.
> 3. Clarification of $\alpha$ notation in Derivations. We appreciate the feedback on the possible confusion of using α and provide an improving suggestion. Indeed, $\alpha$ appears in multiple contexts (in background we simply borrow the notation from previous literature), leading to current ambiguity. To improve clarity, we will modify the notation as suggested:
> Change away the present α introduced above equation (6), making it explicit that the regularization $\alpha$ in section 3.3 is the same as the one in background RLHF.
> Clearly define $E_θ = -\alpha \log(p_θ/p_{ref})$ section 3.3 to 1. Simply the notation and 2. reinforce the connection to DPO modeling.
> 4. Incremental gains of EBA Over EBA-DPO. The reviewer notes that EBA achieves on-par performance with EBA-DPO in table 1. We shall argue that EBA is designed to provide a more structured and interpretable framework rather than just maximizing out the performance in the given setting (Atlas MD data). We also discovered that small energy difference will lead to relatively large weight difference due to the exponential exp, thus making the weighted objective (EBA) resemble the EBA-DPO (binary, win-lose) in this case. More ideal tasks to show advantages of EBA can contain simpler systems with well-balanced energy differences. On the other hand, adapting DPO for protein conformation generation is also novel, and EBA provides a general framework and theoretical justification for what DPO optimizes in this context.
> 5. Presentation and minor corrections (comments). We also appreciate the reviewer’s attention to detail and address the specific presentation issues. We have revised accordingly:
> - Correcting the equation in Line 090 to show the exact transition notation.
> - Clarifying the connection between $x_θ$ and $ϵ_θ$ in the background section.
> - Fixing the notation below equation (14) to ensure $x̂_0$ is properly represented.
> - Adding axis labels and annotations for Figures 3 and 4.
> 6. Additional literature on energy-based models. We appreciate the suggestion to reference broader energy-based modeling literature, such as the mentioned (Song and Kingma, 2021). We will incorporate relevant citations and compare our approach with established energy-based training techniques, including those that require MCMC-based partition function estimation.
>
>
> Finally, we kindly thank the reviewer Gaxj for their insightful feedback, which has helped us refine both methodology and presentation. We will appreciate any further suggestions of revisions in our following discussion.

---

### Decision · Program_Chairs · 2025-05-01

**Decision:**

Accept (poster)

**Comment:**

This paper introduces Energy-Based Alignment (EBA), a method for fine-tuning diffusion models to generate protein conformational ensembles. By incorporating energy-based feedback from molecular simulations into the generative training process, the authors propose a way to bridge data-driven learning and classical physical modeling. Empirical evaluation shows improved performance over prior methods, such as AlphaFlow and MDGen.

The paper addresses an important and timely problem in the intersection of structural biology and generative modeling. The proposed method is well-motivated and represents a promising direction for incorporating physical priors into deep generative models. Several aspects of the paper, including the application to protein ensembles and the connection to preference optimization (EBA-DPO), were recognized as meaningful contributions.

On the flip side, the reviewers raised concerns about the clarity of the presentation. Specifically, the derivation of the EBA objective and the approximations involved in aligning with the Boltzmann distribution were found to be difficult to follow. Questions remained about the rigor of the theoretical framework, particularly regarding the approximation of the partition function and its implications for the learned distribution. Additionally, some reviewers felt that the empirical gains over standard fine-tuning were modest, and requested clearer baselines and more thorough comparisons.

Despite these reservations, the overall direction of the work is promising. The paper introduces a solid contribution to an important area and lays the groundwork for further advances. With additional work to improve clarity, expand theoretical justifications, and benchmark against more recent baselines, this line of research has the potential to make significant impact. On balance, I recommend acceptance.